# Causal-EPIG: Causally Aligned Active CATE Estimation

**Erdun Gao** [1 2]    **Jake Fawkes** [3]    **Dino Sejdinovic** [1 2]

## Abstract

Estimating the Conditional Average Treatment Effect (CATE) is constrained by the high cost of obtaining outcome measurements, making active learning valuable. However, conventional strategies suffer from a fundamental objective mismatch: they reduce uncertainty in model parameters or observable outcomes rather than the unobservable causal quantities of interest. We address this via the principle of causal objective alignment, positing that acquisition functions should target potential outcomes or CATE directly. We operationalize this through Causal-EPIG, a framework adapting Expected Predictive Information Gain to quantify uncertainty reduction in causal quantities. We derive two distinct strategies: a comprehensive approach that targets the joint potential-outcome structure, and a focused approach that directly targets the CATE estimand for sample efficiency. We provide theoretical justification for our framework, establishing a formal link between CATE estimation error and posterior uncertainty in causal quantities. Extensive experiments demonstrate that our strategies improve sample efficiency over standard baselines, and crucially, reveal that the preferred strategy is context-dependent, contingent on the base estimator and treatment-effect structure. Our framework thus provides a principled guide for sample-efficient CATE estimation in practice.

## 1. Introduction

Understanding the causal effects of interventions is central to reliable decision-making in complex domains. Causal inference provides a principled framework for this purpose by modeling the underlying dependencies in real-world data (Pearl, 2009; Hernán & Robins, 2010; Wager, 2024). Its importance is evident across domains such as healthcare (Foster et al., 2011), economics (Heckman, 2000), and recommendation systems (Gao et al., 2024a), where accurately assessing the impact of actions is critical for designing effective policies and delivering personalized interventions. Estimating the Conditional Average Treatment Effect (CATE) is a key problem in this context, as it captures how treatment effects vary across individuals (Künzel et al., 2019). While randomized controlled trials remain the gold standard for causal inference, they are often impractical due to prohibitive costs and ethical barriers (Benson & Hartz, 2017). Consequently, researchers increasingly rely on observational data, which scale more readily but introduce the additional challenge of controlling for confounding to ensure valid causal conclusions (Imbens & Rubin, 2015).

Beyond the challenge of controlling for confounding, a critical practical constraint in observational studies is the acquisition of ground-truth outcome data. This typically requires a costly process, such as expert annotation or long-term patient follow-up, to obtain a reliable outcome for each subject (Nwankwo et al., 2025). Consequently, in many real-world scenarios, this process is expensive, logistically demanding, and subject to privacy or ethical restrictions (Gao et al., 2024b; Kallus & Mao, 2025). In healthcare, for example, measuring outcomes may require costly diagnostic tests or invasive procedures such as biopsies and large-scale tumor imaging, where the resulting label scarcity can severely impact the accuracy of CATE estimation (Bi et al., 2019; Wen et al., 2025). In economics and the social sciences, outcomes such as long-term income trajectories or behavioral changes often require extensive, costly follow-up (McKenzie, 2012). These resource constraints are further compounded when the study population differs systematically from the target population of interest (Kern et al., 2024). For instance, a health maintenance organization in California might need to rely on evidence from a study conducted years prior in Switzerland, whose participants fail to reflect the heterogeneity of the local population (Kallus et al., 2018). This challenge of generalizing findings across populations, formally known as ensuring external validity (Rothwell, 2005) or, more specifically, transportability (Bareinboim & Pearl, 2013), critically undermines the real-world utility of

---

[1]Responsible AI Research Centre, Australian Institute for Machine Learning [2]School of Mathematical Sciences, Adelaide University [3]Department of Statistics, University College London. Correspondence to: Erdun Gao <erdun.gao@adelaide.edu.au>.

*Proceedings of the 43rd International Conference on Machine Learning*, Seoul, South Korea. PMLR 306, 2026. Copyright 2026 by the author(s).

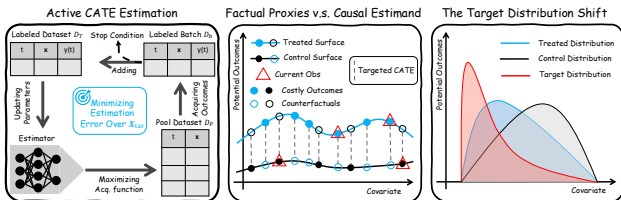

*Figure 1.* **(Left)** Illustrates the pool-based AL pipeline for CATE estimation. **(Middle)** Highlights the proxy-target disconnect: the goal is to learn the CATE, the data consist only of single factual outcomes. **(Right)** Shows the challenge of target distribution shift, where the sampling pool differs from the target population.

causal estimates. To address the dual challenges of resource scarcity and population shift, effective methods should be sample-efficient and target-aware, so that CATE estimates remain accurate for the population of interest.

**Challenges.** Active learning (AL) offers a principled framework for maximizing estimation accuracy under a fixed budget, yet its application to CATE estimation is hindered by a fundamental challenge. Standard AL methods are built for a world of factual observations, designed to reduce uncertainty about observable outcomes or model parameters. The objective of CATE estimation, however, is to precisely quantify an unobservable counterfactual difference. This misalignment between a fact-based acquisition process and a counterfactual-based goal can lead to inefficient data selection, as acquired outcomes may fail to reduce uncertainty where it matters most: in the treatment effect itself.

Existing literature has made valuable progress in adapting conventional AL paradigms for CATE estimation. Seminal works (Jesson et al., 2021) have explored criteria like factual outcome uncertainty or information gain about model parameters. A detailed discussion of related work is provided in App. A. While these criteria are natural and often effective, they remain indirect for CATE estimation: they optimize uncertainty in surrogate quantities rather than in the target causal estimand itself. This issue becomes more pronounced under distribution shift, when the sampling pool may not represent the target population. Consequently, a critical gap persists: *the need for causally aligned acquisition strategies that directly target treatment-effect uncertainty over the population of interest.*

**Contributions.** This paper makes the following contributions. *A New Principle.* We introduce the principle of Causal Objective Alignment (COA), arguing that the structural disconnect between observable data and the causal estimand mandates acquisition functions that are explicitly designed for the final causal goal (Sec. 3). *A Target-Aligned Information-Theoretic Framework.* We develop Causal-EPIG, an information-theoretic framework that operationalizes our principle (Sec. 4.1). From this unified framework, we derive two distinct, principled acquisition strategies: one that targets the joint potential-outcome structure, and a second that directly targets the final CATE estimand. *Broad Model Compatibility.* We demonstrate that Causal-EPIG is a flexible framework that naturally accommodates a range of popular Bayesian CATE estimators (Sec. 4.2), including Gaussian Process (GP)-based models like Causal Multi-task GP (Alaa & Van Der Schaar, 2017) and Non-Stationary GP (Alaa & Schaar, 2018), as well as the tree-based Bayesian Causal Forests (Hahn et al., 2020). *Theoretical Justification.* We provide a formal theoretical justification for our framework (Sec. 4.4), establishing a rigorous link between CATE estimation error and posterior uncertainty in causal quantities (Prop. 4.1), clarifying the relationship between prediction-oriented and parameter-based information targets (Prop. E.4), and providing a convergence analysis that bounds posterior uncertainty under a global greedy acquisition strategy (Thm. 4.3). *Extensive Empirical Validation.* We conduct comprehensive experiments showing that both strategies derived from our framework improve sample efficiency over a wide array of baselines (Sec. 5). Our main experiments and targeted ablations further show that the choice between the comprehensive and focused strategies reflects a context-dependent trade-off, governed by both the base estimator and how the treatment-effect signal is organized relative to the full outcome structure.

## 2. Preliminaries and Problem Setup

**Potential Outcomes and CATE Estimation.** Our analytical framework is grounded in the Neyman-Rubin Potential Outcomes (POs) model (Rubin, 2005). We define the random variables $\mathbf{x}$, $\mathbf{t}$, and $\mathbf{y}$ to represent the covariates, treatment, and outcome, respectively, with domains $\mathcal{X}$, $\{0,1\}$, and $\mathcal{Y}$. We denote realizations by $\boldsymbol{x}$, $t$, and $y$. The two potential outcomes are $\mathbf{y}(0)$ and $\mathbf{y}(1)$, corresponding to the outcome under control and treatment. The propensity score is $\pi(\boldsymbol{x}) = p(\mathbf{t} = 1|\mathbf{x} = \boldsymbol{x})$. Our primary goal is to estimate the CATE, defined as $\tau(\boldsymbol{x}) := \mathbb{E}[\mathbf{y}(1) - \mathbf{y}(0) \mid \mathbf{x} = \boldsymbol{x}]$. To ensure identifiability, we impose the following assumptions.

**Assumption 2.1. Unconfoundedness:** Given the covariates $\mathbf{x}$, treatment assignment $\mathbf{t}$ is independent of the potential outcomes, i.e., $(\mathbf{y}(1), \mathbf{y}(0)) \perp\!\!\!\perp \mathbf{t}|\mathbf{x}$. This implies that $\mathbf{x}$ captures all common causes of treatment and outcome. **Positivity (Common Support):** For any covariates $\boldsymbol{x}$ in the support of the covariate distribution, the probability of receiving either treatment is non-zero: $0 < \pi(\boldsymbol{x}) < 1$. **SUTVA (Stable Unit Treatment Value):** An individual's potential outcomes are unaffected by the treatment assignments of others (No Interference), and the observed outcome is the potential outcome corresponding to the treatment received, i.e., $\mathbf{y} = t\mathbf{y}(1) + (1-t)\mathbf{y}(0)$ (Consistency).

Under Ass. 2.1, the CATE becomes identifiable as the difference in the conditional expectations of the observed outcome, which we denote $f(\boldsymbol{x}, t) := \mathbb{E}[\mathbf{y}|\mathbf{x} = \boldsymbol{x}, \mathbf{t} = t]$. This is

expressed as:

$$\tau(\boldsymbol{x}) = \mathbb{E}[\mathrm{y}|\mathbf{x} = \boldsymbol{x}, \mathrm{t} = 1] - \mathbb{E}[\mathrm{y}|\mathbf{x} = \boldsymbol{x}, \mathrm{t} = 0]. \quad (1)$$

### 2.1. Pool-based active estimation of CATE

We begin with a large unlabeled pool of instances $D_P = \{(\boldsymbol{x}_i, t_i)\}_{i=1}^{n_P}$ and a small initial labeled training set $D_T = \{(\boldsymbol{x}_i, t_i, y_i)\}_{i=1}^{n_T}$. The AL loop proceeds iteratively: a model trained on the current $D_T$ informs an acquisition function, which selects a batch of $n_b$ instances from $D_P$ to be labeled. These are added to $D_T$, and the process repeats until a budget of $n_B$ labels is exhausted (Jesson et al., 2021; Qin et al., 2021). Our objective is to learn a CATE model, $\hat{\tau}(\boldsymbol{x})$, that is accurate over a specific target distribution of interest, $p_{\text{tar}}(\boldsymbol{x})$, which may differ from the distribution of the sampling pool $p_{\text{pool}}(\boldsymbol{x})$. In the standard no-shift setting, this target distribution can coincide with the pool distribution; in transductive settings, it is represented by an unlabeled target covariate set. To formalize this, we evaluate performance using the square root of the Precision in Estimating Heterogeneous Effects ($\sqrt{\epsilon_{\text{PEHE}}}$) (Hill, 2011). This metric is defined as the root mean squared error over the target distribution and is empirically estimated using a finite target set $\boldsymbol{X}_{\text{tar}}$ drawn from $p_{\text{tar}}(\boldsymbol{x})$:

$$\sqrt{\epsilon_{\text{PEHE}}}[\hat{\tau}] := \sqrt{\mathbb{E}_{p_{\text{tar}}(\boldsymbol{x})}\left[(\hat{\tau}(\boldsymbol{x}) - \tau(\boldsymbol{x}))^2\right]}. \quad (2)$$

*Remark* 2.2. **Observational Constraint vs. Experimental Design.** A key constraint in our setup is that we operate on observational data, even during acquisition. For any instance $(\boldsymbol{x}_i, t_i)$, we can only query its pre-existing factual outcome $y_i = \mathrm{y}_i(t_i)$ and cannot intervene to assign a new treatment and observe the counterfactual. This limitation distinguishes our problem from adaptive experimental design, which requires the freedom to assign treatments (Toth et al., 2022; Kato et al., 2024; Cha & Lee, 2025; Klein et al., 2025; Zhang et al., 2025). This constraint is common in sensitive domains like healthcare and social sciences, where treatment assignment is governed by external factors. A more detailed discussion of the related literature on adaptive experimental design is provided in App. A.3.

> ⚑ **Key Objective.** In this problem setup, the central challenge is to design a principled utility function, $U(\cdot)$, that quantifies the informativeness of any single candidate data point $(\boldsymbol{x}, t)$ from the pool. The acquisition strategy is then to select the candidate, denoted $(\boldsymbol{x}_s, t_s)$, that is deemed most valuable by maximizing this function:
>
> $$(\boldsymbol{x}_s, t_s) = \underset{(\boldsymbol{x}, t) \in D_P}{\arg\max} U(\boldsymbol{x}, t \mid D_T, \boldsymbol{X}_{\text{tar}}). \quad (3)$$
>
> While this defines the selection of a single instance, this process is typically extended to the batch setting

by greedily selecting the $n_b$ candidates that yield the highest utility scores.

## 3. Aligning AL with Causal Objectives

This section analyzes the structure of active outcome acquisition for CATE estimation, highlighting an objective mismatch with standard AL paradigms. We show that this mismatch motivates a core principle, Causal Objective Alignment (COA), for designing principled and sample-efficient acquisition strategies.

**From Indirect Proxies to the Causal Estimand.** In standard AL, the path from query to knowledge is direct. The learning objective is aligned with the data-generating process: one queries a point $\boldsymbol{x}_i$ to observe a label $y_i$, which is a direct (though noisy) signal for the target function $f(\boldsymbol{x}_i)$. Direct applications of this paradigm to CATE estimation simply adopt these standard targets: they might focus on the uncertainty of the observable response surface, $f(\boldsymbol{x}, t)$, or on the uncertainty of the model's internal parameters, $\theta$. However, this can create an objective mismatch for CATE estimation, as illustrated in Fig. 1. Both the data we can acquire (factual outcomes) and the model's parameters are indirect proxies for our true inferential goal. This goal is to estimate causal quantities that are not directly observed: the two potential-outcome response surfaces, $f(\boldsymbol{x}, 0)$ and $f(\boldsymbol{x}, 1)$, and the CATE function, $\tau(\boldsymbol{x})$, derived from them as in Eq. 1. This disconnect between proxy acquisition targets and the causal quantities of interest motivates our core design principle:

**Principle 3.1** (Causal Objective Alignment). *An effective acquisition strategy for active CATE estimation should be causally aligned. Its utility function should quantify the value of a query with respect to causal quantities of interest, such as potential-outcome response surfaces or the CATE itself, rather than only indirect proxies such as factual outcomes or model parameters.*

The COA principle also makes the target population explicit: utility is evaluated relative to $\boldsymbol{X}_{\text{tar}}$, the covariates on which CATE accuracy is required. This naturally frames active CATE estimation as a transductive learning problem (a connection detailed in App. A.2.1). It also reveals a spectrum of acquisition strategies, ranging from approaches that target indirect proxies (e.g., factual uncertainty or parameter uncertainty) to causally aligned strategies that target the joint potential-outcome surfaces or the CATE itself. The importance of this alignment is amplified under distribution shift ($p_{\text{tar}}(\boldsymbol{x}) \neq p_{\text{pool}}(\boldsymbol{x})$), where the sampling pool may not represent the target population. This perspective provides the conceptual foundation for the Causal-EPIG framework introduced next.

# 4. Active CATE Estimation via Causal-EPIG

We operationalize the COA principle through the Causal-EPIG framework. We derive two strategies representing a trade-off between comprehensiveness and focus, arguing that the preferred choice depends on the estimator and how the treatment-effect signal is organized. Following their formal definitions, we demonstrate the framework's compatibility with Bayesian CATE models.

## 4.1. The Causal-EPIG Acquisition Function

The design of Causal-EPIG is best motivated by a direct contrast with these standard AL criteria, as illustrated in Tab. 1. As defined above, methods like EIG/BALD are *parameter-focused*, aiming to reduce uncertainty over the model parameters ($\theta$). This objective is indirect; reducing global parameter uncertainty does not guarantee a targeted reduction in CATE uncertainty (Jesson et al., 2021). This limitation can persist even for causal adaptations. For instance, in models like BCF that adopt a separable structure, $f(\boldsymbol{x}, t) = \mu(\boldsymbol{x}) + t \cdot \tau(\boldsymbol{x})$, with parameters $\theta = (\theta_\mu, \theta_\tau)$, one could target the CATE parameters $\theta_\tau$ specifically. This is a more causally aligned parameter-space objective, but it still focuses on the model's internal representation rather than the target causal predictions themselves (Fawkes et al., 2025). In restricted finite-dimensional settings where $\tau(\boldsymbol{x})$ is a direct linear readout of $\theta_\tau$, parameter- and prediction-focused objectives can be closely related; beyond this regime, nonlinear dependence and target-population weighting can induce different acquisition rankings. Standard EPIG (Smith et al., 2023) elevates the objective by targeting future predictions, but in its usual form these predictions remain factual outcomes. This is not fully aligned with CATE, which is inherently a comparative quantity, $\tau(\boldsymbol{x}) = \mathbb{E}[y(1) - y(0) \mid \boldsymbol{x}]$. A data point that is highly informative for one potential outcome may offer little information about the other, and thus may not efficiently reduce uncertainty about their difference.

☛ **A Comprehensive Strategy: Targeting the Joint PO Structure (Causal-EPIG-$\mu$).** A direct application of our COA principle is to target the joint potential-outcome mechanism for a target individual, represented by the joint distribution of $(y^*(0), y^*(1))$. This approach correctly accounts for the inherent dependence between the two outcomes. This leads to our Potential Outcome-based (PO-based) strategy:

$$\text{Causal-EPIG-}\mu(\boldsymbol{x}, t) \coloneqq \mathbb{E}_{p_{\text{tar}}(\boldsymbol{x}^*)} \big[ \mathrm{I}(y; (y^*(0), y^*(1)) \mid \boldsymbol{x}^*, D_T') \big]. \quad (4)$$

In the main text, we use the potential-outcome notation $y^*(0), y^*(1)$ to emphasize the causal targets of interest. In the GP-based theoretical analysis, these targets are instantiated as the latent conditional response surfaces $f(\boldsymbol{x}^*, 0)$ and $f(\boldsymbol{x}^*, 1)$, while $y$ denotes the future factual noisy observation obtained by querying $(\boldsymbol{x}, t)$. By seeking data that reduce uncertainty over this joint distribution, Causal-EPIG-

$\mu$ encourages learning the arm-specific outcome surfaces from which the CATE is derived. This can be advantageous when treatment-effect information is distributed across the two surfaces, but it may also spend acquisition budget on prognostic variation that is less relevant to their contrast. For completeness, we also discuss a simpler, additive variant in App. D.2, which approximates this broader objective.

☛ **A Focused Strategy: Directly Targeting the Causal Estimand (Causal-EPIG-$\tau$).** In contrast, an alternative approach is to focus the acquisition objective directly on the final inferential goal: the CATE function $\tau(\boldsymbol{x}^*)$. This focused strategy can be more sample-efficient when the treatment-effect signal is sufficiently learnable and concentrated in the differenced quantity, by prioritizing data points that directly resolve uncertainty in this causal estimand. Formally, we define the Causal-EPIG-$\tau$ utility as the expected IG about the CATE:

$$\text{Causal-EPIG-}\tau(\boldsymbol{x}, t) \coloneqq \mathbb{E}_{p_{\text{tar}}(\boldsymbol{x}^*)} \big[ \mathrm{I}(y; \tau(\boldsymbol{x}^*) \mid \boldsymbol{x}^*, D_T') \big]. \quad (5)$$

The mutual information represents the expected reduction in CATE posterior entropy. An equivalent and computationally useful formulation uses the KL divergence to frame this utility as the expected belief update about the CATE, $\tau(\boldsymbol{x}^*)$, after a potential observation $y$. Intuitively, a high utility score signifies that an observation at $(\boldsymbol{x}, t)$ is expected to significantly change our beliefs about the CATE in the target population, marking it as a highly informative candidate.

**Positioning and Trade-off.** Causal-EPIG differs from Causal-BALD-style criteria (Jesson et al., 2021) in its information target. Whereas methods such as $\tau$-BALD measure how much a target prediction reveals about model parameters, Causal-EPIG measures how much a future factual observation reveals about target causal predictions. Thus, informativeness is evaluated in prediction space rather than parameter space. Our practical utilities use a mean-marginal aggregation over $X_{\text{tar}}$, estimating the target expectation by a finite sum; a global alternative would compute mutual information with the full vector of target predictions, e.g., $\mathrm{I}(y; \boldsymbol{\tau})$, thereby capturing inter-target dependencies at higher computational cost (Hübotter et al., 2024). Within Causal-EPIG, the $\mu$- and $\tau$-based utilities reflect a comprehensiveness–focus trade-off: Causal-EPIG-$\mu$ preserves the joint potential-outcome structure, while Causal-EPIG-$\tau$ targets the differenced estimand directly. The preferred choice is therefore context-dependent, varying with the estimator and with how treatment-effect information is organized relative to the full outcome structure.

## 4.2. Realization with Bayesian CATE Estimators

Causal-EPIG is estimator-agnostic at the level of the acquisition target, but its implementation requires access to predictive uncertainty over the relevant causal quantities.

*Table 1.* Comparison of information-theoretic acquisition functions for active CATE estimation. For brevity, $D'_T$ denotes the context after conditioning on the candidate covariates and treatment, $D'_T = D_T \cup \{(\boldsymbol{x}, t)\}$, before observing the random outcome $y$.

| | **Non-Causal-Aware** | **Causal-Aware** |
|---|---|---|
| **EIG** | $\mathrm{I}(y; \theta \mid D'_T)$   ($\mu$-BALD) | $\mathrm{I}(y; \theta_\tau \mid D'_T)$   (Causal-EIG) |
| **EPIG** | $\mathbb{E}_{p_{\mathrm{pool}}(\boldsymbol{x}^*, t^*)} \big[ \mathrm{I}(y; y^* \mid (\boldsymbol{x}^*, t^*), D'_T) \big]$ | $\mathbb{E}_{p_{\mathrm{tar}}(\boldsymbol{x}^*)} \big[ \mathrm{I}(y; (y^*(0), y^*(1)) \mid \boldsymbol{x}^*, D'_T) \big]$   (PO-based) 
 $\mathbb{E}_{p_{\mathrm{tar}}(\boldsymbol{x}^*)} \big[ \mathrm{I}(y; \tau(\boldsymbol{x}^*) \mid \boldsymbol{x}^*, D'_T) \big]$   (CATE-based) |

We outline realization strategies for two major classes of Bayesian CATE estimators.

**Exact Realization with GP Models.** For CATE estimators based on GPs, such as CMGP (Alaa & Van Der Schaar, 2017) and NSGP (Alaa & Schaar, 2018), the joint posterior predictive distribution over any set of points is, by construction, a multivariate Gaussian. Consequently, the required predictive variances and covariances can be extracted directly from the GP's analytical posterior covariance matrix. In this setting, the mutual information has an exact closed-form solution. For example, in the scalar jointly Gaussian case,

$$\mathrm{I}(\mathrm{a}; \mathrm{b}) = \frac{1}{2} \log \frac{\mathbb{V}\mathrm{ar}[\mathrm{a}] \mathbb{V}\mathrm{ar}[\mathrm{b}]}{\mathbb{V}\mathrm{ar}[\mathrm{a}] \mathbb{V}\mathrm{ar}[\mathrm{b}] - \mathrm{Cov}[\mathrm{a}, \mathrm{b}]^2}. \quad (6)$$

This allows for an efficient and exact implementation of Causal-EPIG with GP-based models.

**Approximate Realization for General Bayesian Models.** For more complex models whose posterior predictive distribution is represented by samples, such as the MCMC output of Bayesian regression trees (Hill, 2011) and BCF (Hahn et al., 2020), exact mutual information is generally unavailable. To make Causal-EPIG tractable for this broad class of models, we employ a computationally efficient Gaussian approximation, following prior work (Kirsch, 2023; Jesson et al., 2021). This strategy involves forming paired posterior predictive samples of the candidate outcome and the target causal quantity, and then fitting a multivariate Gaussian to these paired draws. The mean vector and covariance matrix are estimated empirically from the set of $n_M$ posterior samples. For instance, when applying this to BCF, the covariance term is computed from its MCMC draws as

$$\mathrm{Cov}[y, \tau(\boldsymbol{x}^*)] \approx \mathrm{Cov}\big( \{f(\boldsymbol{x}, t \mid \theta_j)\}_{j=1}^{n_M}, \{\tau(\boldsymbol{x}^* \mid \theta_j)\}_{j=1}^{n_M} \big). \quad (7)$$

Once this approximation is made, we can reuse the closed-form solution for mutual information (Eq. 6). This approach provides a practical recipe for pairing Causal-EPIG with sample-based estimators while avoiding expensive nested MC simulations.

### 4.3. Budgeted Acquisition Algorithm

The Causal-EPIG utility serves as our acquisition function for ranking candidate data points. We employ this utility in an iterative active learning strategy designed to estimate the CATE under a fixed budget (Qin et al., 2021; Jesson et al., 2021). This process is illustrated conceptually in Fig. 1 and detailed in Alg. 1 (App. C.1). It begins with a warm-start phase where a small random batch of data is labeled. Subsequently, in each round, the algorithm computes the Causal-EPIG utility for all candidates in the unlabeled pool. A batch of points with the highest utility scores is selected, their outcomes are queried, and the CATE model is retrained on the expanded labeled set. This cycle continues until the budget is exhausted. The batch size controls the usual adaptivity–computation trade-off: smaller batches allow more frequent model updates, while larger batches reduce retraining cost.

### 4.4. Theoretical Analysis

We provide a theoretical justification for the Causal-EPIG framework. We first show that, under a Bayesian model-based squared-error criterion, active CATE estimation reduces to minimizing posterior CATE variance. We then analyze posterior-uncertainty convergence for the global PO-based Causal-EPIG objective, and discuss the scope of this guarantee for the practical mean-marginal variant.

#### 4.4.1. OBJECTIVE: CATE ERROR AND POSTERIOR CATE VARIANCE

The following result connects CATE estimation error to a tractable model-based criterion and shows that the relevant uncertainty is the posterior variance of the CATE, which depends on the joint posterior structure of the two potential outcomes. Here, $\epsilon_{\mathrm{PEHE}}^{\mathcal{M}}$ denotes the squared PEHE model risk, not its square root.

**Proposition 4.1.** *Assume $f(\boldsymbol{x}, 0)$ and $f(\boldsymbol{x}, 1)$ are modeled by a joint GP with fixed hyperparameters and Gaussian observation likelihood, and the CATE estimator is the posterior mean $\hat{\tau}_s(\boldsymbol{x}) = \mathbb{E}_s[\tau(\boldsymbol{x})]$. For one-step active CATE estimation under this model-based criterion, minimizing the expected model-based estimation error, $\mathbb{E}_{s+1}\big[\epsilon_{\mathrm{PEHE}}^{\mathcal{M}}(\hat{\tau}_{s+1})\big]$, is equivalent under the GP assumption to minimizing the integrated posterior CATE variance:*

$$\underset{(\boldsymbol{x},t) \in D_P}{\arg\min} \mathbb{E}_{s+1}\big[\epsilon_{\mathrm{PEHE}}^{\mathcal{M}}(\hat{\tau}_{s+1})\big] \equiv \underset{(\boldsymbol{x},t) \in D_P}{\arg\min} \mathbb{E}_{p_{\mathrm{tar}}(\boldsymbol{x}^*)}\big[\mathbb{V}\mathrm{ar}_{s+1}[\tau(\boldsymbol{x}^*)]\big]. \quad (8)$$

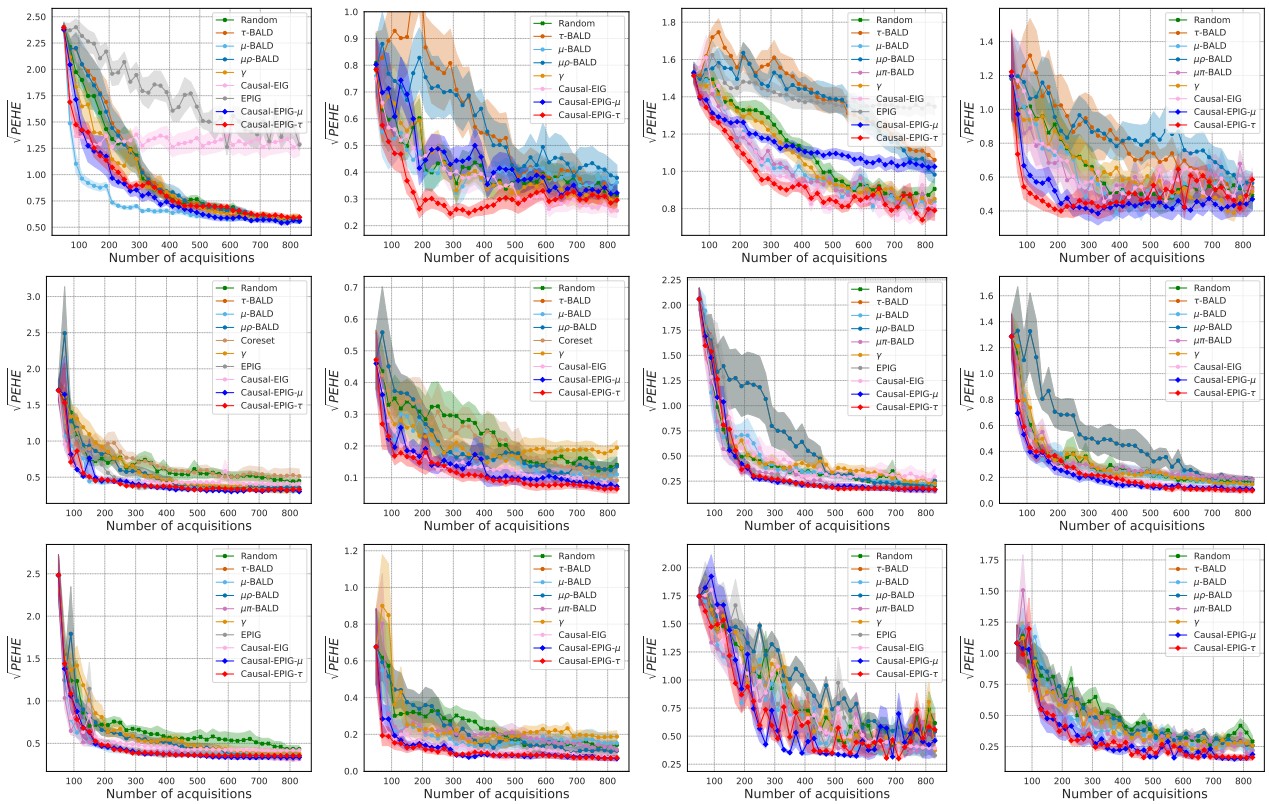

*Figure 2.* Comparison of $\sqrt{\text{PEHE}}$ for BCF, CMGP, and NSGP (rows). The first two columns show results on CausalBALD, while the last two correspond to Hahn, covering both regular and distributional shift settings.

*Here, $\mathbb{V}\text{ar}_{s+1}[\tau(\boldsymbol{x}^*)]$ is the posterior variance of the Bayesian random variable $\tau(\boldsymbol{x}^*)$, which explicitly retains the joint posterior structure:*

$$
\begin{aligned}
\mathbb{V}\text{ar}_{s+1}[\tau(\boldsymbol{x}^*)] = {}& \mathbb{V}\text{ar}_{s+1}[f(\boldsymbol{x}^*,1)] + \mathbb{V}\text{ar}_{s+1}[f(\boldsymbol{x}^*,0)] \\
& - 2\,\text{Cov}_{s+1}\big(f(\boldsymbol{x}^*,1), f(\boldsymbol{x}^*,0)\big).
\end{aligned}
\tag{9}
$$

A detailed proof is provided in App. E.8. This result clarifies why Causal-EPIG should target predictive uncertainty in causal quantities rather than uncertainty in parameters alone. In particular, the CATE variance depends not only on the marginal uncertainties of the two outcome surfaces, but also on their posterior covariance. This motivates acquisition functions that retain the joint potential-outcome structure.

#### 4.4.2. CONVERGENCE OF POSTERIOR UNCERTAINTY

Prop. 4.1 shows that reducing posterior CATE variance is the natural model-based objective. For convergence analysis, we focus on the global PO-based variant, Causal-EPIG-$\mu$-G, which jointly targets all potential outcomes over $\boldsymbol{X}_{\text{tar}}$. We map this objective to transductive active learning (TAL) (Hübotter et al., 2024) by viewing the two potential-outcome surfaces as a single GP $f(\tilde{\boldsymbol{x}})$ on the augmented space $\tilde{\mathcal{X}} = \mathcal{X} \times \{0,1\}$. This induces an augmented target space $\tilde{\boldsymbol{X}}_{\text{tar}}$, containing target potential-outcome components, and an augmented pool space $\tilde{\mathcal{D}}_P$, containing factual

covariate–treatment pairs available for acquisition. Under this mapping, Causal-EPIG-$\mu$-G is exactly Global ITL on the augmented TAL instance (App. E.1).

Let $D_0$ denote any warm-start set and $D_n$ the labeled set after $n$ active acquisitions. All mutual information and variance terms below are computed under the posterior GP after conditioning on $D_0$. Define the information capacity and irreducible uncertainty as $\gamma_{n_B} := \max_{\tilde{A} \subseteq \tilde{\mathcal{D}}_P,\, |\tilde{A}| \leq n_B} \text{I}(f_{\tilde{\boldsymbol{X}}_{\text{tar}}}; \mathbf{y}_{\tilde{A}} \mid D_0)$, $\eta^2_{\tilde{\mathcal{D}}_P}(\tilde{\boldsymbol{x}}^*) := \mathbb{V}\text{ar}[f(\tilde{\boldsymbol{x}}^*) \mid D_0, f_{\tilde{\mathcal{D}}_P}]$. The TAL convergence argument requires a submodularity-type condition on the global set utility. We state it explicitly.

**Assumption 4.2.** The Causal-EPIG-$\mu$-G set utility $\psi_{\tilde{\boldsymbol{X}}_{\text{tar}}}(\tilde{A}) := \text{I}(f_{\tilde{\boldsymbol{X}}_{\text{tar}}}; \mathbf{y}_{\tilde{A}} \mid D_0)$, $\tilde{A} \subseteq \tilde{\mathcal{D}}_P$, is submodular.

This assumption is used only for the global PO-based convergence analysis. It should not be read as a universal submodularity claim for all Causal-EPIG variants; in fully transductive settings, or for derived-target and mean-marginal objectives, synergistic effects may occur. We discuss its scope in App. E.2.

**Theorem 4.3.** *Suppose data acquisition follows greedy Causal-EPIG-$\mu$-G, and let $n_B$ denote the total number of actively acquired outcomes. Under standard GP assumptions, Ass. 4.2, and the information-capacity regularity condition*

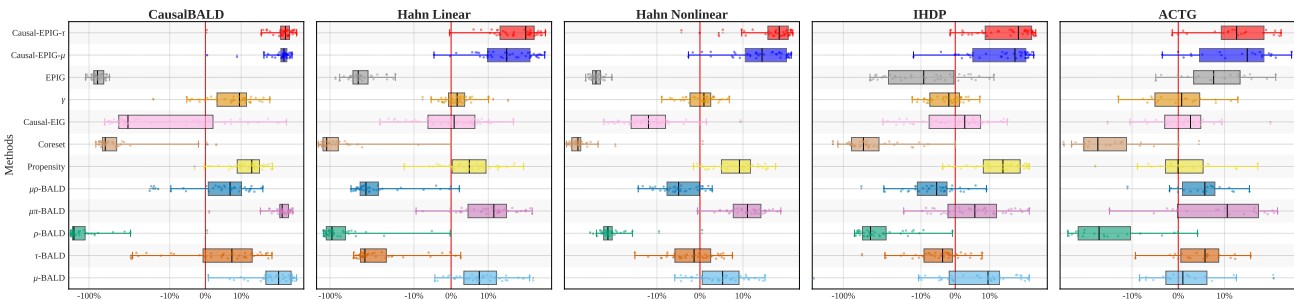

*Figure 3.* Average relative improvement of acquisition functions over *Random acquisition* on five datasets.

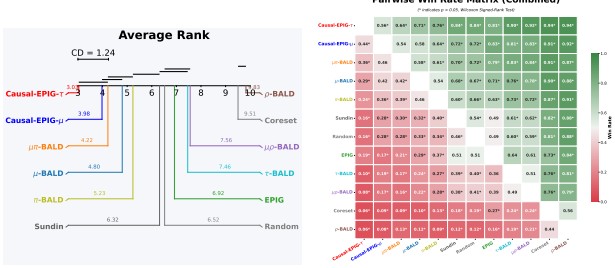

*Figure 4.* Average comparison of acquisition functions across all synthetic datasets: (a) critical difference diagram, and (b) pairwise win-rate matrix with statistical significance.

*required by* Hübotter et al. *(2024, Thm. 3.3), there exists* $C > 0$ *such that, for any* $1 \leq n_B \leq |\tilde{\mathcal{D}}_P|$ *and any* $\tilde{\boldsymbol{x}}^* \in \tilde{\boldsymbol{X}}_{\text{tar}}$,

$$\mathbb{V}\mathrm{ar}[f(\tilde{\boldsymbol{x}}^*) \mid D_{n_B}] \leq \eta^2_{\mathcal{D}_P}(\tilde{\boldsymbol{x}}^*) + C\frac{\gamma_{n_B}}{\sqrt{n_B}}. \quad (10)$$

The proof reduces Causal-EPIG-$\mu$-G to Global ITL on the augmented TAL instance and applies Hübotter et al. (2024, Thm. 3.3) (App. E.3). The result bounds target posterior uncertainty by an irreducible floor $\eta^2_{\mathcal{D}_P}$ plus a capacity-controlled term. In our finite pool-based setting, the theorem should be read as a finite-budget posterior-uncertainty bound. In the idealized TAL limit with increasing or repeatable sample access, if $\gamma_{n_B}/\sqrt{n_B} \to 0$, the bound converges to the irreducible uncertainty floor. It is a convergence characterization under the stated assumptions, not a universal optimality guarantee.

**Mean-marginal objective.** The global objective is theoretically convenient but computationally expensive. Our experiments therefore use the mean-marginal utilities in Eqs. 4–5. These do not exploit all inter-target dependencies and are not covered directly by the global theorem. However, for the $\tau$-based mean-marginal utility, a relaxed TAL-style argument under an additional monotonicity condition gives an average-variance analogue: $\frac{1}{|\boldsymbol{X}_{\text{tar}}|} \sum_{\boldsymbol{x}^* \in \boldsymbol{X}_{\text{tar}}} \mathbb{V}\mathrm{ar}[\tau(\boldsymbol{x}^*) \mid D_{n_B}] \lesssim \frac{1}{|\boldsymbol{X}_{\text{tar}}|} \sum_{\boldsymbol{x}^* \in \boldsymbol{X}_{\text{tar}}} \eta^2_{\mathcal{D}_P, \tau}(\boldsymbol{x}^*) + C\frac{\bar{\gamma}^\tau_{n_B}}{\sqrt{n_B}}$. For the PO-based mean-marginal variant, marginal PO uncertainty provides a sufficient, though potentially loose, route to controlling marginal CATE uncertainty; a formal vector-valued

MM bound would require additional analysis. Details are provided in App. E.5.

# 5. Experimental Results

To assess the sample efficiency of our Causal-EPIG framework, we conduct extensive experiments on several benchmarks. These include synthetic datasets based on the data-generating processes from Causal-BALD (Jesson et al., 2021) and Hahn et al. (2020), as well as two well-established semi-synthetic benchmarks: the Infant Health and Development Program (IHDP) (Hill, 2011) and AIDS Clinical Trials Group Study 175 (ACTG-175) (Hammer et al., 1996). Full details regarding the dataset characteristics and partitioning for each benchmark are available in App. B.

**Base Bayesian CATE Estimators, Baselines, and Metrics.** To demonstrate the flexibility of our framework, we implement Causal-EPIG with three distinct and well-established Bayesian CATE estimators: BCF, CMGP, and NSGP. These models are natural partners for our information-theoretic acquisition functions, as they provide the necessary posterior uncertainty over the relevant causal quantities. For brevity, the main text focuses on these primary models; comprehensive results for all setups, including an additional estimator from the Causal-BALD study, are provided in App. F. For our baselines, we compare against a range of acquisition functions, including Random, $\gamma$-acquisition (S-type error rate control) (Sundin et al., 2019), coreset selection (Qin et al., 2021), Causal-EIG (Fawkes et al., 2025), and the suite of methods from Causal-BALD (Jesson et al., 2021). Detailed implementations for all methods are available in App. C. Our primary evaluation metric is the Root PEHE ($\sqrt{\hat{\epsilon}_{\text{PEHE}}}$ or $\sqrt{\text{PEHE}}$ for short; Eq. 2), computed on the target set $\boldsymbol{X}_{\text{tar}}$. All results are reported as the mean and standard deviation across 10 independent runs. In addition to performance curves, we report the relative Root PEHE improvement over the Random baseline for a holistic summary of sample efficiency. This metric is calculated at each acquisition step $k$ as $(\sqrt{\text{PEHE}}_{\text{Random}}(k) - \sqrt{\text{PEHE}}_{\text{Method}}(k))/\sqrt{\text{PEHE}}_{\text{Random}}(k)$. Finally, we aggregate these point-wise improvements across all steps to visualize the distribution of performance gains for each method,

offering insight into its consistency throughout the active learning process.

## 5.1. Synthetic Data

**Results.** Fig. 2 shows that Causal-EPIG improves sample efficiency on synthetic benchmarks. The focused strategy, **Causal-EPIG-$\tau$** (red), achieves leading or competitive sample efficiency across base estimators (BCF, CMGP, NSGP) and distribution-shift settings. The comprehensive strategy, **Causal-EPIG-$\mu$** (blue), also significantly outperforms most baselines, though its effectiveness is slightly attenuated with BCF. One possible explanation is BCF's separation of prognostic ($\mu$) and treatment ($\tau$) effects, which may make joint potential-outcome targeting less direct. Fig. 3 shows that Causal-EPIG-$\tau$ yields the highest average improvement, supporting the efficiency of direct targeting when the CATE signal is well captured by the estimator. Statistical analysis of 180 dataset-seed combinations (Fig. 4) corroborates these findings. In the Critical Difference (CD) diagram, Causal-EPIG-$\tau$ achieves the best average rank (3.03), followed by Causal-EPIG-$\mu$ (3.98); both rank ahead of the remaining baselines under the Nemenyi comparison (Nemenyi test, CD=1.24, $\alpha = 0.05$). Pairwise Wilcoxon signed-rank tests further confirm that Causal-EPIG-$\tau$ significantly outperforms 11 of 11 baselines ($p < 0.05$) with an 80.6% win rate, while Causal-EPIG-$\mu$ beats 8 baselines with a 72.1% win rate. Notably, neither variant significantly loses to any baseline. Comprehensive results and ablation studies on stability (varying initializations, pool sizes, batch sizes, and the Deep-GP estimator) are provided in App. F.1, F.2.

**Mechanistic comparison of Causal-EPIG-$\mu$ and Causal-EPIG-$\tau$.** To clarify the trade-off between the two Causal-EPIG targets, we run a targeted synthetic study varying the treatment-effect formation mechanism and the predictive share of the active informative block, $\rho = \frac{k_{\mathrm{pred}}}{k_{\mathrm{prog}}+k_{\mathrm{pred}}}$, while keeping the total active informative dimension fixed, following the predictive/prognostic composition used in related active feature-acquisition studies (Piskorz et al., 2025). Causal-EPIG-$\mu$ preserves the full joint potential-outcome structure, whereas Causal-EPIG-$\tau$ compresses the target to the differenced quantity. Thus, the relevant factor is not merely the number of predictive variables, but how the treatment-effect signal is organized relative to the full outcome structure. Fig. 5 shows two regimes. In the *surface-split* setting, the effect is induced by differences between arm-specific outcome surfaces, so learning the full PO pair remains beneficial and Causal-EPIG-$\mu$ is favored. In the *cancellation* settings, part of the full-outcome structure cancels or becomes weakly relevant in the treatment-effect difference, making direct CATE targeting more efficient; the relative advantage therefore shifts toward Causal-EPIG-$\tau$ as $\rho$ increases.

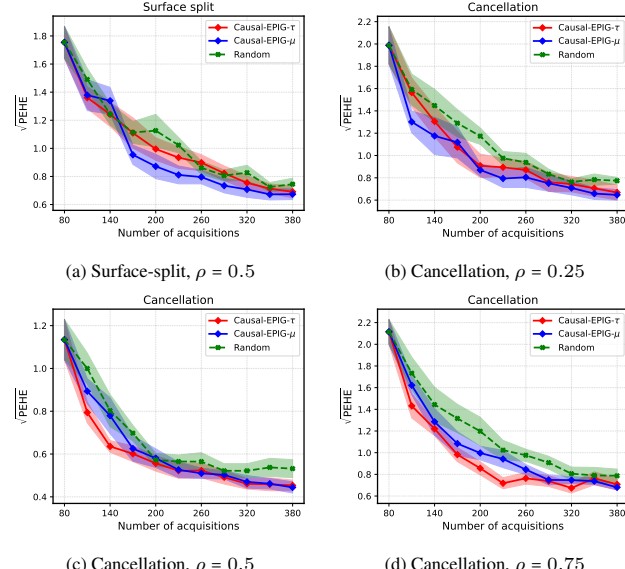

*Figure 5.* Targeted synthetic comparison of Causal-EPIG-$\mu$ and Causal-EPIG-$\tau$. Surface-split favors Causal-EPIG-$\mu$, while cancellation regimes increasingly favor Causal-EPIG-$\tau$ as the predictive share $\rho$ grows.

**Computational Considerations.** The sample efficiency of Causal-EPIG comes at the cost of a more computationally intensive acquisition function than simpler baselines. This is a standard computational–statistical trade-off. All main experiments use the mean-marginal formulation, whose acquisition cost scales linearly with the pool size, target-set size, and number of predictive samples. The approach is therefore most suitable when outcome acquisition is expensive relative to model scoring, such as in clinical or industrial studies. In such settings, the additional computation can be justified by improved sample efficiency. A detailed breakdown of per-sample runtimes is provided in App. D.3.

## 5.2. Semi-synthetic Data

**Benchmarks.** We evaluate our framework on two well-established semi-synthetic benchmarks designed to mimic challenges of real-world observational studies. The first, the IHDP (Hill, 2011), simulates selection bias by removing a non-random subset of the treated group from a randomized trial. The second, ACTG-175 (Hammer et al., 1996), constructs an observational cohort by excluding participants based on their enrollment symptoms.

**Results and Analysis.** The results on these semi-synthetic benchmarks (Fig. 6 and Fig. 3) further support the practical effectiveness of Causal-EPIG and illustrate the trade-off between its two strategies. Across both the IHDP and ACTG datasets, both Causal-EPIG-$\mu$ and Causal-EPIG-$\tau$ achieve competitive or leading performance, demonstrating the overall strength of our causally-aligned, prediction-focused approach. The strong performance of the compre-

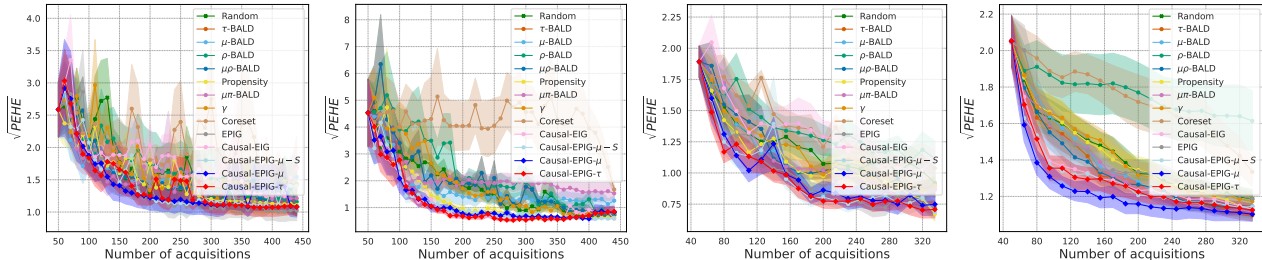

*Figure 6.* Comparison of $\sqrt{\text{PEHE}}$ on IHDP and ACTG-175 datasets. We evaluate the following four experimental settings: (1) CMGP on IHDP, (2) CMGP on IHDP under target shift, (3) CMGP on ACTG-175, and (4) BCF on ACTG-175.

hensive Causal-EPIG-$\mu$ strategy is particularly noteworthy, suggesting that its more comprehensive targeting of the joint potential-outcome structure is highly effective in these complex, lower signal-to-noise settings. Furthermore, these results underscore our thesis that the optimal acquisition strategy is context-dependent. On the IHDP benchmark, which is defined by significant selection bias, we observe that propensity-based baselines also perform competitively. This is consistent with IHDP's strong selection-bias structure, where propensity-aware criteria are naturally competitive. Overall, the key advantage of Causal-EPIG is not that one variant is universally dominant, but that the framework provides two target-aligned acquisition rules whose relative strengths vary with the estimator and data-generating structure. Full results are provided in App. F.1.

## 6. Discussions

This work addressed the objective mismatch between standard active learning and CATE estimation by introducing the principle of causal objective alignment. We operationalized this principle with the Causal-EPIG framework, a flexible information-theoretic approach that yields two distinct acquisition strategies: a comprehensive strategy targeting the joint potential-outcome structure and a focused strategy targeting the CATE itself. Our experiments show that both target-aligned strategies generally improve sample efficiency over strong baselines, and further support our central hypothesis that the choice between them reflects a context-dependent trade-off. This finding provides empirical evidence for our principle: aligning the acquisition objective with the causal target is important, but the preferred form of alignment depends on the estimator and data-generating structure. By providing a framework that navigates this trade-off, our work supports more cost-effective and reliable CATE estimation where outcome acquisition is expensive.

**Limitations and Future Work.** Causal-EPIG assumes no unobserved confounding and relies on the base estimator's predictive uncertainty, especially its joint uncertainty over the relevant causal quantities. This uncertainty may be unreliable in very low-data regimes or under severe model misspecification (Zhang et al., 2025). While estimator-agnostic

at the acquisition-target level, its effectiveness is bounded by the underlying CATE estimator and by the quality of its uncertainty estimates. Future work includes integrating robust models such as CausalPFN (Balazadeh et al., 2025), addressing hidden confounding (Li et al., 2023), and extending the framework to adaptive experimental design.

## Acknowledgments

This work is supported by the Responsible AI Research Centre (RAIR). EG is also supported by the Centre for Augmented Reasoning (CAR).

## Impact Statement

This work aims to make CATE estimation more sample-efficient when outcome acquisition is costly, with potential benefits in resource-constrained domains such as personalized medicine and development economics. By accounting for target populations, Causal-EPIG may also help improve external validity under population shift. Its deployment, however, still depends on valid causal assumptions: reducing CATE uncertainty does not ensure correctness under model misspecification or unobserved confounding.

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

# Appendix

## Table of Contents

# A. Additional Related Works

## A.1. Active CATE Estimation

Our work addresses *active outcome acquisition for CATE estimation*: treatment assignments are observational, but outcome measurements are costly (Nwankwo et al., 2025). The goal is to select which outcomes to query from an existing cohort in order to improve a CATE model most efficiently. Existing approaches have mainly adapted standard active learning heuristics. One line uses diversity-based sampling, such as coreset selection, to cover the covariate space (Qin et al., 2021; Wen et al., 2025). Another targets specific causal error types rather than the overall CATE estimation error (Sundin et al., 2019). These methods are useful, but their acquisition objectives remain indirect, relying on geometric diversity or auxiliary error criteria rather than uncertainty in the target causal estimand. More closely related are information-theoretic causal acquisition rules. Causal-EIG (Fawkes et al., 2025) targets information gain about CATE-specific parameters ($\theta_\tau$), while Causal-BALD (Jesson et al., 2021) measures how much a causal prediction informs the model parameters. More broadly, Gao & Sejdinovic (2026) study active estimation of general causal quantities beyond CATE, deriving acquisition rules from posterior uncertainty of causal functionals. These works are causal-aware, but they either operate through parameter-space objectives or target broader causal functionals rather than the CATE-specific predictive quantities considered here. In contrast, Causal-EPIG builds on prediction-oriented Bayesian active learning (Smith et al., 2023) and defines informativeness directly with respect to target causal predictions, such as potential outcomes or the CATE itself. This yields acquisition rules aligned with the causal estimand over the target population, rather than only with geometric coverage, auxiliary error criteria, or parameter uncertainty.

## A.2. Inductive and Transductive Goals in Active Learning

Active learning is often framed as an inductive problem: the goal is to train a model that generalizes to future unseen data under an IID assumption (Settles, 2009). Transductive active learning instead optimizes performance on a specified target set. Pool-based AL has elements of both: acquisition decisions are made using a finite unlabeled pool, but the usual goal is still to learn a broadly generalizable model. This distinction becomes important under distribution shift, where the sampling pool distribution $p_{\text{pool}}$ may differ from the target distribution $p_{\text{tar}}$. Recent target-aware acquisition strategies formalize this transductive perspective (MacKay, 1992; Hübotter et al., 2024; Smith et al., 2023).

### A.2.1. CONNECTION BETWEEN ACTIVE CATE ESTIMATION AND TAL

Active CATE estimation can be viewed as a transductive problem because acquisition should be evaluated with respect to a target population $\boldsymbol{X}_{\text{tar}}$. In TAL (Hübotter et al., 2024), the learner queries from a sample space $\mathcal{S}$ to reduce uncertainty on a target space $\mathcal{A}$. In our setting, $\boldsymbol{X}_{\text{tar}}$ plays the role of the target set: the goal is not uniform accuracy over the whole covariate space, but accurate CATE estimation on this population. The CATE setting also has an additional structural feature. A query at $(\boldsymbol{x}_i, t_i)$ reveals only one factual outcome, providing information about the corresponding response surface $f(\boldsymbol{x}_i, t_i)$. The inferential target, however, is the contrast $\tau(\boldsymbol{x}_i) = f(\boldsymbol{x}_i, 1) - f(\boldsymbol{x}_i, 0)$, which depends on a pair of potential-outcome response surfaces. Thus, even when $p_{\text{pool}}(\boldsymbol{x}) = p_{\text{tar}}(\boldsymbol{x})$, acquisition remains target-dependent: the utility of a factual observation should be judged by how much it reduces uncertainty in the unobserved causal quantities over $\boldsymbol{X}_{\text{tar}}$. Under distribution shift, this transductive aspect is further strengthened because the learner must use observations from $p_{\text{pool}}$ to improve CATE estimates on $p_{\text{tar}}$. This perspective motivates the Causal Objective Alignment principle in Sec. 3 and the Causal-EPIG framework in Sec. 4.

## A.3. Adaptive Experimental Design

A separate line of causal active learning studies active or adaptive experimental design, where the learner can influence treatment assignment. One major direction studies adaptive randomization, where treatment probabilities are updated to improve efficiency for estimators such as the ATE (van der Laan, 2008; Hahn et al., 2011). Recent work refines these designs using online nuisance estimates, martingale arguments, and specialized estimators for adaptively collected data (Kato et al., 2021; Tabord-Meehan, 2023; Kato et al., 2020; Cook et al., 2024). A second direction studies finite-pool design, including classical covariate-balance trade-offs (Efron, 1971) and modern sampling methods based on leverage scores or balancing walks (Addanki et al., 2022; Ghadiri et al., 2023; Harshaw et al., 2024). A third direction uses Bayesian experimental design for treatment-effect estimation, including methods with regulatory constraints (Klein et al., 2025), structural uncertainty from causal discovery (Toth et al., 2022), GP-based CATE variance minimization (Cha & Lee, 2025), finite-sample guarantees in network settings (Zhang et al., 2025), and designs that use observational data as a prior for prospective trials (Gao et al.,

2026). Our setting is different. Adaptive experimental design asks *who should receive which treatment*; active outcome acquisition asks *whose outcome should be measured*, given that treatment assignments are already fixed by observational or practical constraints. This distinction is important in domains such as healthcare, where treatments may already have been assigned but outcome measurements, such as biopsies or genetic sequencing, remain costly. Nevertheless, both settings allocate limited resources to reduce causal uncertainty, suggesting that target-aware acquisition principles may also inform future adaptive experimental design.

## B. Datasets

We evaluate Causal-EPIG on two fully synthetic benchmarks and two semi-synthetic benchmarks based on IHDP and ACTG-175. In all settings, the outcome-generation process is known, so the ground-truth CATE is available for PEHE evaluation. We also include covariate-shift variants for the CausalBALD, Hahn, and IHDP datasets. **General protocol.** For synthetic benchmarks, we generate a pool set $D_P$ of 2000 instances, a validation set of 200 instances, and a test set of 2000 instances. Unless otherwise stated, active learning starts from 50 randomly labeled instances. For CausalBALD and Hahn, we run 40 acquisition rounds with batch size 20; for IHDP, 40 rounds with batch size 10; and for ACTG-175, 20 rounds with batch size 15. Results are averaged over 10 independent trials. In the standard setting, the acquisition target set is the pool covariates, $X_{\text{tar}} = X_P$, and evaluation is reported on both pool and held-out test covariates. In the covariate-shift setting, the acquisition target set is the shifted test set, $X_{\text{tar}} = X_{\text{test}}$, and test performance is the primary metric.

### B.1. CausalBALD Synthetic Dataset

We use a one-dimensional synthetic dataset adapted from CausalBALD (Jesson et al., 2021), which itself builds on Kallus et al. (2019). In the standard setting, $x \sim \mathcal{N}(0,1)$, $t \mid x \sim \text{Bern}(\pi(x))$, and $\pi(x) = \text{sigmoid}(2x + 0.5)$. The potential-outcome means are $\mu_0(x) = 1 + 2\sin(2x)$ and $\mu_1(x) = 2x + 3 - 2\sin(2x)$, with additive noise $\epsilon \sim \mathcal{N}(0,1)$. Hence, $\tau(x) = \mu_1(x) - \mu_0(x) = 2x + 2 - 4\sin(2x)$. For the covariate-shift setting, training and pool covariates remain drawn from $\mathcal{N}(0,1)$, while test/target covariates are drawn from $\mathcal{U}(0.2, 0.5)$. The outcome functions are unchanged.

### B.2. Hahn Synthetic Dataset

We use the simulation design from Hahn et al. (2020), with covariates $x \in \mathbb{R}^5$. The first three covariates are Gaussian, $x_1, x_2, x_3 \sim \mathcal{N}(0,1)$; $x_4 \sim \text{Bernoulli}(0.5)$; and $x_5 \sim \mathcal{U}\{1,2,3\}$. We use the nonlinear prognostic function and heterogeneous treatment-effect function: $\mu(x) = -6 + g(x_5) + 6|x_3 - 1|$ and $\tau(x) = 1 + 2x_2x_4$, where $g(1) = 2$, $g(2) = -1$, and $g(3) = -4$. We use a modified propensity score to create a more challenging observational setting. Let $\tilde{\mu}(x) = 3\mu(x)/\sigma_\mu$, where $\sigma_\mu$ is the population standard deviation of $\mu(x)$. We define $\pi(x) = 0.8\,\phi(\tilde{\mu}(x)) - 0.5x_1 + \xi$, with $\xi \sim \mathcal{U}(0.05, 0.15)$, and draw $t \sim \text{Bernoulli}(\pi(x))$. The observed outcome is $y = \mu(x) + t\tau(x) + \epsilon$, with Gaussian noise scaled to signal-to-noise ratio 3. For the covariate-shift setting, the pool is generated as above, while the test continuous covariates are drawn from $x_1, x_2, x_3 \sim \mathcal{U}(0.2, 0.5)$. The discrete covariates and outcome functions remain unchanged.

### B.3. IHDP Semi-Synthetic Dataset

We use the Infant Health and Development Program (IHDP) semi-synthetic benchmark (Hill, 2011). The dataset contains 747 subjects, with 139 treated and 608 controls, and 25 covariates: 6 continuous and 19 binary. We split the data into 523 pool/training subjects and 224 test subjects, and standardize continuous covariates. In the standard setting, a sparse coefficient vector $\beta_B$ is generated by sampling each entry from $\{0.0, 0.1, 0.2, 0.3, 0.4\}$ with probabilities $\{0.6, 0.1, 0.1, 0.1, 0.1\}$. The potential-outcome means are $\mu_0(x) = \exp((x + 0.5)\beta_B)$ and $\mu_1(x) = (x + 0.5)\beta_B - \omega_B$, where $\omega_B$ is chosen so that the ATT equals 4. Potential outcomes are generated as $y_0(x) = \mu_0(x) + \epsilon$ and $y_1(x) = \mu_1(x) + \epsilon$, with $\epsilon \sim \mathcal{N}(0,1)$. For the covariate-shift setting, the first two continuous test covariates, birth weight and head circumference, are resampled from $\mathcal{U}(0, 0.5)$. The coefficient vector is sampled as above, but the first two coefficients are set to zero. The outcome functions are $\mu_0(x) = \exp((x + 0.5)\beta_B)$ and $\mu_1(x) = \exp((x + 0.5)\beta_B) + 3x_{\text{bw}}x_{\text{b.head}}$, so that $\tau(x) = 3x_{\text{bw}}x_{\text{b.head}}$.

### B.4. ACTG-175 Semi-Synthetic Dataset

We use the AIDS Clinical Trials Group Study 175 (ACTG-175) dataset (Hammer et al., 1996). Following the semi-synthetic setup, an observational cohort is formed from the randomized trial by removing a non-random subset of patients, specifically those not showing symptomatic HIV infection. The resulting dataset contains 813 subjects, with 281 treated and 532

---

**Algorithm 1** Budgeted Batch Active Learning for CATE Estimation

---

**Require:** Unlabeled pool $D_P$, target set $\boldsymbol{X}_{\text{tar}}$, utility function $U$, batch size $n_b$, max budget $n_B$.
**Ensure:** Final labeled set $D_T$ and final CATE estimator $\hat{\tau}(\cdot)$.

1: Initialize labeled set $D_T \leftarrow \varnothing$.
   // – Warm-start Phase –
2: Select an initial random batch $D_{\text{init}} \subset D_P$.
3: Query factual outcomes and form labeled batch $\widetilde{D}_{\text{init}} = \{(\boldsymbol{x}, t, y) : (\boldsymbol{x}, t) \in D_{\text{init}}\}$.
4: Update $D_T \leftarrow D_T \cup \widetilde{D}_{\text{init}}$ and $D_P \leftarrow D_P \setminus D_{\text{init}}$.
5: Train initial CATE estimator $\hat{\tau}(\cdot)$ on $D_T$.
   // – Main Active Learning Loop –
6: **while** $|D_T| < n_B$ **and** $D_P \neq \varnothing$ **do**
7:     Compute $U(\boldsymbol{x}_i, t_i \mid D_T, \boldsymbol{X}_{\text{tar}})$ for each $(\boldsymbol{x}_i, t_i) \in D_P$.
8:     Select a batch $D_b$ of $n_b$ candidates according to the acquisition rule.
9:     Query factual outcomes for all $(\boldsymbol{x}, t) \in D_b$.
10:    Update $D_T \leftarrow D_T \cup D_b$ and $D_P \leftarrow D_P \setminus D_b$.
11:    Retrain or update $\hat{\tau}(\cdot)$ on $D_T$.
12: **end while**
13: **return** $D_T, \hat{\tau}(\cdot)$.

---

controls, and 12 covariates: 3 continuous and 9 binary. We split the data into 569 pool/training subjects and 244 test subjects, standardize continuous covariates, and simulate outcomes using nonlinear functions with interactions. The prognostic score and CATE are

$$\mu(x) = 6 + 0.3 x_{\text{wtkg}}^2 - \sin(x_{\text{age}})(x_{\text{gender}} + 1) + 0.6 x_{\text{hemo}} x_{\text{race}} - 0.2 x_{\text{z30}},$$
$$\tau(x) = 1 + 1.5 \sin(x_{\text{wtkg}})(x_{\text{karnof\_hi}} + 1) + 2 x_{\text{age}}. \tag{11}$$

We set $\mu_0(x) = \mu(x)$ and $\mu_1(x) = \mu(x) + \tau(x)$. Potential outcomes are generated as $y_0(x) = \mu_0(x) + \epsilon$ and $y_1(x) = \mu_1(x) + \epsilon$, where $\epsilon \sim \mathcal{N}(0, \sigma_y^2)$ and $\sigma_y = \frac{\max(\mu) - \min(\mu)}{8}$. The observed outcome is $y = (1 - t) y_0(x) + t y_1(x)$.

## C. Model and Acquisition Details

This section summarizes the active CATE acquisition loop, the acquisition functions used in our experiments, and the base CATE estimators. Additional hyperparameter details are provided in the released code.

### C.1. Active CATE Estimation Loop

Alg. 1 describes the batch active-learning protocol used throughout our experiments. Starting from a random warm-start set, each round scores candidates in the remaining pool, acquires a batch of factual outcomes, augments the labeled set, and retrains or updates the CATE estimator.

**Batch acquisition.** For all acquisition functions, candidates are first scored individually. For baselines adapted from Causal-BALD (Jesson et al., 2021), we follow the original stochastic softmax acquisition protocol. For Causal-EPIG, we use deterministic top-$n_b$ selection, which was empirically comparable to temperature-based stochastic variants while being simpler and faster.

### C.2. Acquisition Functions

We compare Causal-EPIG with several active acquisition baselines. Unless otherwise stated, batch selection is performed by selecting the $n_b$ candidates with the highest utility scores.

**Random.** Random acquisition selects $n_b$ candidates uniformly at random from the remaining pool $D_P$.

**Causal-BALD variants.** We include the acquisition functions from Causal-BALD (Jesson et al., 2021), which adapt BALD to causal quantities through parameter-space information gains. The main variants are:

- $\tau$-**BALD**: mutual information between the CATE-related quantity and model parameters, e.g., $\mathrm{I}(\mathrm{y}(1) - \mathrm{y}(0); \theta \mid D_T')$.

- $\mu$-**BALD**: mutual information between the relevant potential outcome and model parameters, e.g., $\mathrm{I}(\mathrm{y}(t); \theta \mid D_T \cup \{(\boldsymbol{x}, t)\})$.
- **Propensity**: an acquisition rule targeting uncertainty in the propensity score $\pi(\boldsymbol{x})$.
- **Combined variants**: weighted combinations such as $\mu\pi$-BALD and $\mu\rho$-BALD.

The key distinction is that Causal-BALD is parameter-focused, whereas Causal-EPIG evaluates informativeness in the space of target causal predictions. We use the official implementation from the authors.[1]

**Sign-ambiguity BALD.** Following Sundin et al. (Sundin et al., 2019) and Causal-BALD (Jesson et al., 2021), this baseline targets points where the sign of the CATE is most uncertain. Given posterior CATE samples $\{\tau_k(\boldsymbol{x})\}_{k=1}^K$, let $\sigma_\tau(\boldsymbol{x}) = \mathrm{std}(\{\tau_k(\boldsymbol{x})\}_{k=1}^K)$ and

$$\gamma_k(\boldsymbol{x}) \coloneqq \Phi\left(-\frac{|\tau_k(\boldsymbol{x})|}{\sigma_\tau(\boldsymbol{x})}\right), \tag{12}$$

where $\Phi$ is the standard normal CDF. The utility is approximated as

$$U_{\mathrm{sign}}(\boldsymbol{x}) \coloneqq \mathrm{H}(\mathrm{Bernoulli}(\bar{\gamma}(\boldsymbol{x}))) - \frac{1}{K}\sum_{k=1}^K \mathrm{H}(\mathrm{Bernoulli}(\gamma_k(\boldsymbol{x}))), \tag{13}$$

where $\bar{\gamma}(\boldsymbol{x}) = K^{-1}\sum_{k=1}^K \gamma_k(\boldsymbol{x})$.

**Coreset selection (QHTE).** We implement the coreset acquisition strategy from QHTE (Qin et al., 2021).[2] The pool is partitioned into treatment arms, $D_P^1$ and $D_P^0$, and coreset scores are computed separately within each arm. For a candidate $(\boldsymbol{x}, t)$, the utility is

$$U_{\mathrm{QHTE}}(\boldsymbol{x}, t) \coloneqq \min_{\boldsymbol{x}' \in \boldsymbol{X}_T^t} d(\boldsymbol{x}, \boldsymbol{x}'), \qquad t \in \{0, 1\}, \tag{14}$$

where $d(\cdot, \cdot)$ is derived from posterior predictive covariance when available. Scores from the two arms are then combined to select the top batch.

**Causal-EIG.** Causal-EIG (Fawkes et al., 2025) was proposed for prospective causal effect estimation, where it scores the information gain that a prospective dataset provides about causal parameters. We adapt it to pool-based AL by treating each candidate $(\boldsymbol{x}, t)$ as a size-one prospective dataset:

$$U_{\mathrm{Causal\text{-}EIG}}(\boldsymbol{x}, t) \coloneqq \mathrm{I}(y; \theta_\tau \mid \boldsymbol{x}, t, D_T). \tag{15}$$

We implement this acquisition function with BCF and CMGP using the official code.[3]

**EPIG.** EPIG (Smith et al., 2023) measures how informative a candidate label is about future labels. In our setting, the factual EPIG baseline is

$$U_{\mathrm{EPIG}}(\boldsymbol{x}, t) \coloneqq \mathbb{E}_{p_{\mathrm{tar}}(\boldsymbol{x}^*, t^*)}\left[\mathrm{I}(y; y^*(t^*) \mid (\boldsymbol{x}^*, t^*), D_T')\right]. \tag{16}$$

Unlike Causal-EPIG, this target remains a factual prediction rather than a causal estimand.

**Causal-EPIG.** Our proposed Causal-EPIG utilities replace the factual prediction target in EPIG with causal quantities. Causal-EPIG-$\mu$ targets the joint potential-outcome pair $(y^*(0), y^*(1))$, while Causal-EPIG-$\tau$ targets the CATE $\tau(\boldsymbol{x}^*) = y^*(1) - y^*(0)$. The main definitions are given in Sec. 4.

### C.3. Base CATE Estimators

**Bayesian Causal Forests.** BCF models the conditional outcome as $\mathbb{E}[y \mid \boldsymbol{x}, t] = \mu(\boldsymbol{x}) + t\tau(\boldsymbol{x})$ using Bayesian tree ensembles (Hahn et al., 2020). We use its accelerated extension, XBCF (Krantsevich et al., 2023), with the reparameterization

$$f_\theta(\boldsymbol{x}, t) = a\,\tilde{\mu}_{\mathrm{bcf}}(\boldsymbol{x}) + b_t\,\tilde{\tau}_{\mathrm{bcf}}(\boldsymbol{x}), \tag{17}$$

so that the CATE is

$$\tau(\boldsymbol{x}) = (b_1 - b_0)\tilde{\tau}_{\mathrm{bcf}}(\boldsymbol{x}). \tag{18}$$

---

[1] https://github.com/OATML/causal-bald
[2] https://github.com/Qcer17/QHTE
[3] https://github.com/LucileTerminassian/causal_prospective_merge

Posterior CATE samples are obtained from MCMC draws. For information-based acquisition, we require joint predictive samples of candidate outcomes and target causal quantities. For each posterior draw $s$, we compute paired samples such as $(f^{(s)}(\boldsymbol{x},t), \tau^{(s)}(\boldsymbol{x}^*))$ or $(f^{(s)}(\boldsymbol{x},t), y^{*(s)}(0), y^{*(s)}(1))$, and approximate their joint distribution with a multivariate Gaussian estimated from the empirical mean and covariance of the samples (Kirsch, 2023; Jesson et al., 2021).

**Causal Multitask Gaussian Processes.** CMGP models the two potential outcome functions jointly as a multitask GP (Alaa & Van Der Schaar, 2017). It places a GP prior on $[f_0(\boldsymbol{x}), f_1(\boldsymbol{x})]^\top$ using a matrix-valued kernel, typically constructed through a Linear Model of Coregionalization. Given Gaussian likelihoods, the posterior over the potential outcomes is Gaussian. The CATE posterior is therefore also Gaussian:

$$\tau(\boldsymbol{x}_*) \sim \mathcal{N}\big(\boldsymbol{e}^\top \boldsymbol{\mu}_{\text{post}}(\boldsymbol{x}_*), \boldsymbol{e}^\top \boldsymbol{\Sigma}_{\text{post}}(\boldsymbol{x}_*, \boldsymbol{x}_*)\boldsymbol{e}\big), \qquad \boldsymbol{e} = [-1, 1]^\top. \tag{19}$$

Thus, the mutual information quantities required by Causal-EPIG can be computed directly from the GP posterior covariance.

**Non-Stationary Gaussian Processes.** NSGP defines a GP over the augmented input space $\mathcal{X} \times \{0, 1\}$, treating treatment as an input (Alaa & Schaar, 2018). The model uses a non-stationary kernel allowing the control and treatment surfaces to have different smoothness properties. Posterior inference for $\tau(\boldsymbol{x}_*) = f(\boldsymbol{x}_*, 1) - f(\boldsymbol{x}_*, 0)$ follows from the joint Gaussian posterior over the two potential outcome values. As with CMGP, Causal-EPIG utilities are computed from the analytical posterior covariance.

**Deep Kernel Learning estimator.** For ablation, we also include the DUE estimator used in Causal-BALD (Jesson et al., 2021). DUE combines neural feature learning with a sparse variational GP. We use it to test whether Causal-EPIG remains effective beyond the main BCF/GP estimators.

### C.4. Gaussian Approximation Diagnostics for BCF

For BCF, Causal-EPIG uses a Gaussian approximation to the joint posterior samples needed for mutual information computation. To assess this approximation, we test the joint posterior of $(\hat{y}, \hat{\tau})$ across five DGPs: CausalBALD, Hahn linear, Hahn nonlinear, IHDP, and ACTG. As expected, formal tests reject exact multivariate normality in all settings, but the magnitude of deviation varies. The semi-synthetic datasets show more moderate deviations, with Mardia kurtosis values closer to the bivariate Gaussian value of 8, while the synthetic datasets show heavier tails. We therefore view the Gaussian approximation as a tractable covariance-based approximation rather than an exact posterior assumption.

*Table 2.* Multivariate normality diagnostics for the joint posterior of $(\hat{y}, \hat{\tau})$ under BCF.

| Dataset | Q-Q corr | $\chi^2$ GoF p-value | HZ stat | Mardia kurtosis |
|---|---|---|---|---|
| CausalBALD | 0.974 | $< 10^{-10}$ | 0.210 | 11.430 |
| Hahn (linear) | 0.981 | $< 10^{-10}$ | 0.189 | 9.655 |
| Hahn (nonlinear) | 0.993 | $< 10^{-10}$ | 0.194 | 9.878 |
| IHDP | 0.989 | $1.45 \times 10^{-3}$ | 0.173 | 8.868 |
| ACTG | 0.989 | $6.94 \times 10^{-3}$ | 0.167 | 8.407 |

## D. Interpretations and Derivations

This section provides derivations and additional interpretation for the information-theoretic acquisition functions used in the paper. We focus on Causal-EPIG-$\tau$, since it is the most direct estimand-targeted objective. The derivation for Causal-EPIG-$\mu$ is identical except that the target variable is the joint potential-outcome vector $(y^*(0), y^*(1))^\top$ rather than the scalar CATE.

### D.1. Derivation and Estimation of Causal-EPIG

**Information-gain form.** For a candidate query $(\boldsymbol{x}, t)$ and target covariate $\boldsymbol{x}^*$, let $y$ be the unobserved factual outcome and let $\tau(\boldsymbol{x}^*)$ be the target CATE. The information gain from observing $y$ is the posterior entropy reduction

$$\text{IG}_\tau((\boldsymbol{x}, t), y, \boldsymbol{x}^*) = \text{H}(\tau(\boldsymbol{x}^*) \mid D_T) - \text{H}(\tau(\boldsymbol{x}^*) \mid D_T \cup \{(\boldsymbol{x}, t, y)\}). \tag{20}$$

Averaging over the unknown outcome and target covariates gives

$$\text{Causal-EPIG}_\tau(\boldsymbol{x}, t) \coloneqq \mathbb{E}_{p_{\text{tar}}(\boldsymbol{x}^*)} \mathbb{E}_{p(y \mid \boldsymbol{x}, t, D_T)} \big[\text{IG}_\tau((\boldsymbol{x}, t), y, \boldsymbol{x}^*)\big] \tag{21}$$

$$= \mathbb{E}_{p_{\mathrm{tar}}(\boldsymbol{x}^*)} \mathbb{E}_{p(y,\tau(\boldsymbol{x}^*)|\boldsymbol{x},t,D_T)} \left[ \log \frac{p(\tau(\boldsymbol{x}^*) \mid D_T, y, \boldsymbol{x}, t)}{p(\tau(\boldsymbol{x}^*) \mid D_T)} \right] \tag{22}$$

$$= \mathbb{E}_{p_{\mathrm{tar}}(\boldsymbol{x}^*)} \mathbb{E}_{p(y,\tau(\boldsymbol{x}^*)|\boldsymbol{x},t,D_T)} \left[ \log \frac{p(y,\tau(\boldsymbol{x}^*) \mid \boldsymbol{x}, t, D_T)}{p(y \mid \boldsymbol{x}, t, D_T) p(\tau(\boldsymbol{x}^*) \mid D_T)} \right] \tag{23}$$

$$= \mathbb{E}_{p_{\mathrm{tar}}(\boldsymbol{x}^*)} \left[ \mathrm{I}(y; \tau(\boldsymbol{x}^*) \mid (\boldsymbol{x}, t), D_T) \right]. \tag{24}$$

This is the definition used in Eq. 5. Equivalently, Causal-EPIG can be interpreted as the expected KL divergence between the posterior and prior predictive distributions of the target causal quantity after observing the candidate outcome.

**Model-specific realization.** For GP-based CATE estimators such as CMGP and NSGP, the required joint predictive distributions are Gaussian, so the mutual information terms can be computed analytically from posterior variances and covariances. For sample-based estimators such as BCF, we form paired posterior samples of the candidate outcome and target causal quantity, fit a Gaussian using empirical moments, and then use the same closed-form MI expression. This avoids nested Monte Carlo while retaining the covariance structure needed by the acquisition utility.

### D.1.1. GAUSSIAN CLOSED FORM

Assume that, conditional on $D_T$, the candidate outcome and target CATE have a bivariate Gaussian posterior

$$(y, \tau(\boldsymbol{x}^*)) \mid \boldsymbol{x}, t, \boldsymbol{x}^*, D_T \sim \mathcal{N}(\boldsymbol{\mu}, \boldsymbol{\Sigma}), \tag{25}$$

with

$$\boldsymbol{\Sigma} = \begin{pmatrix} \mathbb{V}\mathrm{ar}[y] & \mathrm{Cov}[y, \tau(\boldsymbol{x}^*)] \\ \mathrm{Cov}[\tau(\boldsymbol{x}^*), y] & \mathbb{V}\mathrm{ar}[\tau(\boldsymbol{x}^*)] \end{pmatrix}. \tag{26}$$

Using the Gaussian entropy formula, $\mathrm{H}(\mathbf{z}) = \frac{1}{2} \log \left( (2\pi e)^d |\boldsymbol{\Sigma}_\mathbf{z}| \right)$, the mutual information is

$$\mathrm{I}(y; \tau(\boldsymbol{x}^*)) = \mathrm{H}(y) + \mathrm{H}(\tau(\boldsymbol{x}^*)) - \mathrm{H}(y, \tau(\boldsymbol{x}^*)) \tag{27}$$

$$= \frac{1}{2} \log \left( \frac{\mathbb{V}\mathrm{ar}[y] \mathbb{V}\mathrm{ar}[\tau(\boldsymbol{x}^*)]}{\det(\boldsymbol{\Sigma})} \right) \tag{28}$$

$$= \frac{1}{2} \log \left( \frac{\mathbb{V}\mathrm{ar}[y] \mathbb{V}\mathrm{ar}[\tau(\boldsymbol{x}^*)]}{\mathbb{V}\mathrm{ar}[y] \mathbb{V}\mathrm{ar}[\tau(\boldsymbol{x}^*)] - \mathrm{Cov}[y, \tau(\boldsymbol{x}^*)]^2} \right). \tag{29}$$

Thus, for a finite target set $\boldsymbol{X}_{\mathrm{tar}}$,

$$\mathrm{Causal\text{-}EPIG}_\tau(\boldsymbol{x}, t) \approx \frac{1}{|\boldsymbol{X}_{\mathrm{tar}}|} \sum_{\boldsymbol{x}^* \in \boldsymbol{X}_{\mathrm{tar}}} \frac{1}{2} \log \left( \frac{\mathbb{V}\mathrm{ar}[y] \mathbb{V}\mathrm{ar}[\tau(\boldsymbol{x}^*)]}{\mathbb{V}\mathrm{ar}[y] \mathbb{V}\mathrm{ar}[\tau(\boldsymbol{x}^*)] - \mathrm{Cov}[y, \tau(\boldsymbol{x}^*)]^2} \right), \tag{30}$$

where the variances and covariance are computed separately for each candidate–target pair.

### D.2. A Taxonomy of Information-Theoretic Acquisition Functions

Causal-EPIG is part of a broader family of information-theoretic acquisition functions. Tab. 3 summarizes the main distinctions: parameter- vs. prediction-targeted objectives, factual vs. causal prediction targets, and mean-marginal vs. global formulations.

The first distinction is the information target. EIG-style methods, including Causal-EIG and Causal-BALD, are parameter-focused: they reduce uncertainty in $\theta_\tau$ or $\theta$. EPIG/ITL-style methods are prediction-focused: they reduce uncertainty in model outputs. Causal-EPIG belongs to the latter family, but replaces factual prediction targets with causal targets. Within prediction-focused methods, factual EPIG targets future factual outcomes, PO-based Causal-EPIG targets the potential-outcome components, and CATE-based Causal-EPIG targets the final estimand $\tau(\boldsymbol{x}^*)$. The PO target can be formulated additively, $\mathrm{I}(y; y^*(0)) + \mathrm{I}(y; y^*(1))$, or jointly, $\mathrm{I}(y; (y^*(0), y^*(1)))$. The joint formulation retains the covariance between potential outcomes and is the primary Causal-EPIG-$\mu$ objective used in the paper; the additive version is denoted Causal-EPIG-$\mu$-S in the appendix experiments. Finally, information can be aggregated over the target population in two ways. The mean-marginal formulation averages pointwise information gains over $\boldsymbol{X}_{\mathrm{tar}}$ and scales well computationally. The global formulation measures information about the full target vector jointly, capturing inter-target dependencies but incurring substantially higher cost. Our main experiments use the mean-marginal formulation for scalability, while global variants are used for theoretical comparison and ablation.

*Table 3.* A taxonomy of information-theoretic acquisition functions for active CATE estimation. The table distinguishes methods by their target of information gain and by whether they use a mean-marginal or global formulation.

| Family | Target of Information Gain | Mean-Marginal Formulation | Global / Full Formulation |
|---|---|---|---|
| **EIG** | Model Parameters $(\theta)$ | $\mathrm{I}(y; \theta_\tau \mid (\boldsymbol{x}, t), D_T)$ $\quad$ or $\quad$ $\mathrm{I}(y; \theta \mid (\boldsymbol{x}, t), D_T)$ | |
| **EPIG / ITL** | Factual Outcome $(y^*)$ | $\mathbb{E}_{p_{\text{pool}}(\boldsymbol{x}^*, t^*)}[\mathrm{I}(y; y^* \mid (\boldsymbol{x}, t), (\boldsymbol{x}^*, t^*), D_T)]$ | $\mathrm{I}(y; \boldsymbol{y}^* \mid (\boldsymbol{x}, t), D_T)$ |
| | Potential Outcomes $(y^*(t^*))$ | $\mathbb{E}_{p_{\text{tar}}(\boldsymbol{x}^*)}\left[\sum_{t^*} \mathrm{I}(y; y^*(t^*))\right]$ (Additive) $\quad$ $\mathbb{E}_{p_{\text{tar}}(\boldsymbol{x}^*)}[\mathrm{I}(y; (y^*(0), y^*(1)))]$ **(Joint PO)** | $\mathrm{I}(y; \boldsymbol{y}_{PO}^* \mid (\boldsymbol{x}, t), D_T)$ |
| | **CATE** $(\tau(\boldsymbol{x}^*))$ | $\mathbb{E}_{p_{\text{tar}}(\boldsymbol{x}^*)}[\mathrm{I}(y; \tau(\boldsymbol{x}^*))]$ | $\mathrm{I}(y; \boldsymbol{\tau} \mid (\boldsymbol{x}, t), D_T)$ |

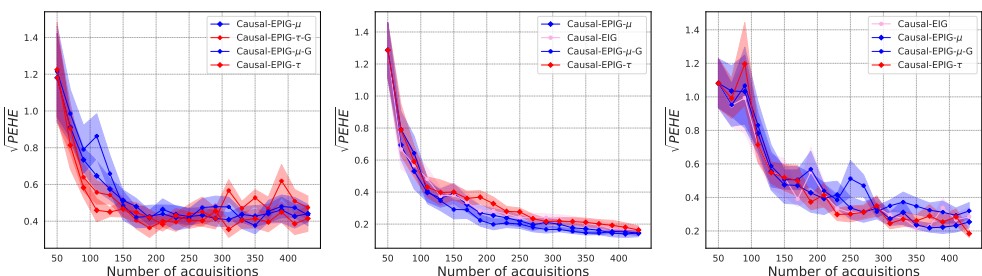

*Figure 7.* Performance comparison on the Hahn linear dataset under distribution shift. Panels show different base CATE estimators. The scalable mean-marginal variants generally match or improve over their global counterparts in this setting.

### D.3. Computational Complexity and Runtime Analysis

The main computational cost of Causal-EPIG is determined by the pool size $n_P$, target-set size $n_{\text{tar}} = |\boldsymbol{X}_{\text{tar}}|$, and number of posterior samples $S$ for sample-based estimators. The mean-marginal formulation used in our experiments scores all candidates in $\mathcal{O}(n_P n_{\text{tar}} S)$, scaling linearly in both pool and target-set size. By contrast, the global formulation requires operations on the full target covariance matrix for each candidate, leading to roughly $\mathcal{O}(n_P n_{\text{tar}}^3)$ cost and making it expensive for large target sets. Tab. 4 reports the empirical acquisition time per batch for mean-marginal and global variants. The mean-marginal objectives are substantially faster, with Causal-EPIG-$\mu$ being about 20× faster than its global version on average. Tab. 5 compares Causal-EPIG-$\tau$ with standard baselines. Causal-EPIG is more expensive than simpler BALD-style baselines, but this reflects the intended computational–statistical trade-off: the method spends additional computation to target uncertainty in the causal estimand, which is appropriate when outcome acquisition is the dominant cost.

*Table 4.* Average acquisition time (seconds) per batch for mean-marginal and global (-G) Causal-EPIG formulations. Results are mean ± std across trials.

| Estimator | Causal-EPIG-$\mu$ | Causal-EPIG-$\mu$-G | Causal-EPIG-$\tau$ | Causal-EPIG-$\tau$-G |
|---|---|---|---|---|
| BCF | $0.7731 \pm 0.0508$ | $16.1392 \pm 0.5763$ | $0.5392 \pm 0.0342$ | $4.1811 \pm 0.1036$ |
| CMGP | $0.1813 \pm 0.0164$ | $4.1814 \pm 0.1598$ | $0.4016 \pm 0.0289$ | $1.9840 \pm 0.0883$ |
| NSGP | $0.3931 \pm 0.0331$ | $7.5014 \pm 1.0600$ | $0.3812 \pm 0.0334$ | $2.9213 \pm 0.4922$ |
| Overall | $0.4492 \pm 0.2999$ | $9.2740 \pm 6.1728$ | $0.4406 \pm 0.0860$ | $3.0288 \pm 1.1025$ |

*Table 5.* Average acquisition time (seconds) of Causal-EPIG-$\tau$ compared with representative baselines.

| Methods | Random | $\mu$-BALD | $\mu\rho$-BALD | $\mu\pi$-BALD | Causal-EPIG-$\tau$ |
|---|---|---|---|---|---|
| | $(6.5 \pm 0.7)$ | $(3.6 \pm 0.1)$ | $(9.6 \pm 0.4)$ | $(9.6 \pm 0.4)$ | $(4.4 \pm 0.1)$ |
| Time (s) | $\times 10^{-5}$ | $\times 10^{-3}$ | $\times 10^{-3}$ | $\times 10^{-3}$ | $\times 10^{-1}$ |

# E. Theoretical Analysis Details

## E.1. Framework Mapping to TAL

Our convergence analysis follows by reducing the global PO-based Causal-EPIG objective to the Global Information-Theoretic Learning (ITL) objective in the transductive active learning (TAL) framework of Hübotter et al. (2024). We make this mapping explicit. Throughout this theoretical analysis, the potential-outcome targets are the latent conditional response surfaces $f(\boldsymbol{x}, t) = \mathbb{E}[Y(t) \mid \boldsymbol{x}]$, rather than additional noisy potential-outcome realizations.

**Augmented GP.** We model the two potential outcome surfaces $f_t(\boldsymbol{x})$ as a single GP $f(\tilde{\boldsymbol{x}})$ over the augmented input space $\tilde{\mathcal{X}} = \mathcal{X} \times \{0, 1\}$, with $\tilde{\boldsymbol{x}} = (\boldsymbol{x}, t)$. The augmented kernel $\tilde{k}$ can be written using a Linear Model of Coregionalization (LMC) construction,

$$\tilde{k}((\boldsymbol{x}, t), (\boldsymbol{x}', t')) \coloneqq \sum_{q=1}^{Q} (B_q)_{t,t'} \, k_q(\boldsymbol{x}, \boldsymbol{x}'), \tag{31}$$

where each $k_q$ is a positive semidefinite kernel and each $B_q \geq 0$ is a $2 \times 2$ coregionalization matrix. Hence $\tilde{k}$ is a valid LMC kernel and induces a joint GP over all potential-outcome components. In the CMGP experiments, we use $Q = 2$, following the CMGP specification.

**Target and sample spaces.** The TAL target space is the augmented set of target potential outcomes, $\mathcal{A} \coloneqq \tilde{\boldsymbol{X}}_{\text{tar}} = \{(\boldsymbol{x}^*, t) : \boldsymbol{x}^* \in \boldsymbol{X}_{\text{tar}}, \ t \in \{0, 1\}\}$. The TAL sample space is the set of factual covariate–treatment pairs available for acquisition, $\mathcal{S} \coloneqq \tilde{\mathcal{D}}_P = \{(\boldsymbol{x}, t) : (\boldsymbol{x}, t) \in \mathcal{D}_P\}$. Thus, target variables are latent potential-outcome components in $\mathcal{A}$, while admissible queries are factual observations in $\mathcal{S}$. In the pool-based acquisition loop, candidates already queried are removed from the remaining pool. We write $\tilde{\mathcal{D}}_P^{(n)} = \tilde{\mathcal{D}}_P \smallsetminus \{\tilde{\boldsymbol{x}}_1, \ldots, \tilde{\boldsymbol{x}}_n\}$ for the remaining augmented pool after $n$ acquisitions. Capacity definitions below are taken over the full accessible sample space $\tilde{\mathcal{D}}_P$.

**Equivalence to Global ITL.** Let $D_0$ denote any warm-start labeled set, and let $D_n$ denote the labeled set after $n$ active acquisitions. Conditioning on $D_0$, the global PO-based version of our acquisition rule, denoted Causal-EPIG-$\mu$-G, selects

$$\tilde{\boldsymbol{x}}_{n+1} \in \underset{\tilde{\boldsymbol{x}} \in \tilde{\mathcal{D}}_P^{(n)}}{\arg\max} \, \mathrm{I}\big(f_{\tilde{\boldsymbol{X}}_{\text{tar}}}; \mathrm{y}_{\tilde{\boldsymbol{x}}} \mid D_n\big). \tag{32}$$

This is the Global ITL rule of Hübotter et al. (2024, Eq. 2) applied to the augmented TAL instance $(\mathcal{A}, \mathcal{S}) = (\tilde{\boldsymbol{X}}_{\text{tar}}, \tilde{\mathcal{D}}_P)$, with the additional pool-based restriction that each factual unit is queried at most once. Therefore, for budgets $n \leq |\tilde{\mathcal{D}}_P|$, the finite-budget convergence characterization for Causal-EPIG-$\mu$-G follows by applying the TAL convergence theorem to this augmented GP. The theoretical implications of using the structured LMC kernel are discussed in Sec. E.6.

## E.2. Submodularity: Scope of the Assumption

The TAL convergence theorem requires a submodularity-type condition on the uncertainty-reduction set function. In our notation, the relevant set function is

$$\psi_{\tilde{\boldsymbol{X}}_{\text{tar}}}(\tilde{A}) \coloneqq \mathrm{I}\big(f_{\tilde{\boldsymbol{X}}_{\text{tar}}}; \mathrm{y}_{\tilde{A}} \mid D_0\big), \qquad \tilde{A} \subseteq \tilde{\mathcal{D}}_P, \tag{33}$$

where the conditioning on the warm-start set $D_0$ is suppressed in the main text by treating the posterior after warm start as the prior for acquisition. The corresponding marginal gain is

$$\Delta_{\tilde{\boldsymbol{X}}_{\text{tar}}}(\tilde{\boldsymbol{x}} \mid \tilde{A}) \coloneqq \psi_{\tilde{\boldsymbol{X}}_{\text{tar}}}(\tilde{A} \cup \{\tilde{\boldsymbol{x}}\}) - \psi_{\tilde{\boldsymbol{X}}_{\text{tar}}}(\tilde{A}). \tag{34}$$

Assumption 4.2 states that this set function is submodular, i.e., $\Delta_{\tilde{\boldsymbol{X}}_{\text{tar}}}(\tilde{\boldsymbol{x}} \mid \tilde{A}') \geq \Delta_{\tilde{\boldsymbol{X}}_{\text{tar}}}(\tilde{\boldsymbol{x}} \mid \tilde{A})$ for all $\tilde{A}' \subseteq \tilde{A} \subseteq \tilde{\mathcal{D}}_P$ and $\tilde{\boldsymbol{x}} \in \tilde{\mathcal{D}}_P \smallsetminus \tilde{A}$.

**When the assumption holds.** For the global PO-based objective considered in Thm. 4.3, a sufficient condition is that the augmented sample space is contained in the augmented target space, $\mathcal{S} \subseteq \mathcal{A}$. In this case, the objective is a standard Gaussian ITL objective over target variables, and submodularity follows from Hübotter et al. (2024, Lemma C.9). In our CATE setting, this sufficient condition holds in a no-shift transductive setting where every factual query component in $\tilde{\mathcal{D}}_P$ is included in the augmented target set $\tilde{\boldsymbol{X}}_{\mathrm{tar}}$.

**When the assumption may fail.** The assumption is not guaranteed in the fully transductive regime $\mathcal{S} \nsubseteq \mathcal{A}$, for example under target covariate shift, where query points need not be components of the target set. It may also fail for derived-target objectives, such as CATE-targeted utilities involving $\tau(\boldsymbol{x}) = f(\boldsymbol{x}, 1) - f(\boldsymbol{x}, 0)$, or for mean-marginal approximations that no longer coincide with the global ITL objective. In such cases, synergistic effects can occur, as in Hübotter et al. (2024, Example C.8). Thus, Assumption 4.2 should be read as a technical condition for the global PO-based convergence analysis, not as a universal submodularity claim for all Causal-EPIG variants.

### E.3. Proof of Thm. 4.3

We prove the result by reduction to Hübotter et al. (2024, Thm. 3.3). Condition on any warm-start set $D_0$ and regard the resulting posterior GP as the prior for the active acquisition process. Let $D_n$ denote the labeled set after $n$ active acquisitions. All mutual informations and variances in this proof are computed under this warm-start posterior.

Under the augmented GP construction in App. E.1, the global PO-based Causal-EPIG-$\mu$-G acquisition rule is

$$\tilde{\boldsymbol{x}}_{n+1} \in \underset{\tilde{\boldsymbol{x}} \in \tilde{\mathcal{D}}_P^{(n)}}{\arg \max} \mathrm{I}\big(f_{\tilde{\boldsymbol{X}}_{\mathrm{tar}}}; \mathrm{y}_{\tilde{\boldsymbol{x}}} \mid D_n\big). \tag{35}$$

This is the Global ITL rule in the TAL problem with target space $\mathcal{A} = \tilde{\boldsymbol{X}}_{\mathrm{tar}}$ and sample space $\mathcal{S} = \tilde{\mathcal{D}}_P$, with the no-replacement restriction induced by our finite pool. Define the TAL information capacity for this augmented problem as

$$\gamma_n := \max_{\tilde{B} \subseteq \tilde{\mathcal{D}}_P, \, |\tilde{B}| \leq n} \mathrm{I}\big(f_{\tilde{\boldsymbol{X}}_{\mathrm{tar}}}; \mathrm{y}_{\tilde{B}} \mid D_0\big). \tag{36}$$

This is the global information capacity appearing in Thm. 4.3, with the warm-start posterior treated as the prior. Also define the irreducible uncertainty for a target potential-outcome component $\tilde{\boldsymbol{x}}^* \in \tilde{\boldsymbol{X}}_{\mathrm{tar}}$ as

$$\eta_{\mathcal{D}_P}^2(\tilde{\boldsymbol{x}}^*) := \mathbb{Var}\big[f(\tilde{\boldsymbol{x}}^*) \mid D_0, f_{\tilde{\mathcal{D}}_P}\big], \tag{37}$$

i.e., the residual posterior uncertainty after observing the entire accessible sample-space latent function in the noiseless limit.

The assumptions required by Hübotter et al. (2024, Thm. 3.3) are imposed by our standard GP observation model, Assumption 4.2, and the stated information-capacity regularity condition: the augmented potential-outcome process is a GP with valid kernel $\tilde{k}$, observations are noisy evaluations of $f(\tilde{\boldsymbol{x}})$ with independent Gaussian noise, the global ITL uncertainty-reduction objective is assumed submodular, and the augmented information capacity satisfies the required regularity condition. Applying Hübotter et al. (2024, Thm. 3.3) to the augmented TAL instance $(\mathcal{A}, \mathcal{S}) = (\tilde{\boldsymbol{X}}_{\mathrm{tar}}, \tilde{\mathcal{D}}_P)$ yields that there exists a constant $C > 0$ such that, for any $1 \leq n \leq |\tilde{\mathcal{D}}_P|$ and any target potential-outcome component $\tilde{\boldsymbol{x}}^* \in \tilde{\boldsymbol{X}}_{\mathrm{tar}}$,

$$\mathbb{Var}[f(\tilde{\boldsymbol{x}}^*) \mid D_n] \leq \eta_{\mathcal{D}_P}^2(\tilde{\boldsymbol{x}}^*) + C \frac{\gamma_n}{\sqrt{n}}. \tag{38}$$

Setting $n = n_B$ gives the finite-budget statement of Thm. 4.3. $\qquad \square$

### E.4. Interpretation of Thm. 4.3

Thm. 4.3 provides a finite-budget convergence characterization of the global Causal-EPIG-$\mu$-G strategy in our pool-based setting. It bounds posterior uncertainty over target potential outcomes by the irreducible uncertainty floor $\eta_{\mathcal{D}_P}^2$ plus a capacity-controlled term. Because our pool-based acquisition queries each factual unit at most once, the theorem should be interpreted for budgets $n \leq |\tilde{\mathcal{D}}_P|$. The asymptotic convergence-to-floor interpretation, i.e., the case where $\gamma_n/\sqrt{n} \to 0$, corresponds to the idealized TAL setting with increasing or repeatable sample access, or to a capacity-controlled limiting characterization beyond the finite pool. This result should not be interpreted as universal optimality over all acquisition objectives. Rather, it establishes that the globally aligned PO-based objective directly controls uncertainty in the target causal quantities under the stated GP, submodularity, and capacity assumptions. The appearance of the capacity term $\gamma_n$ reflects the intrinsic difficulty of transferring information from observable factual outcomes in $\tilde{\mathcal{D}}_P$ to unobserved potential outcomes in $\tilde{\boldsymbol{X}}_{\mathrm{tar}}$.

### E.5. A Relaxed TAL-Style Analysis for Mean-Marginal Causal-EPIG

The analysis above establishes a convergence bound for the *global* joint-PO strategy Causal-EPIG-$\mu$-G by directly mapping it to the Global ITL framework of Hübotter et al. (2024). However, for computational scalability, the strategy used in practice is often the *mean-marginal* (MM) formulation. We therefore provide a relaxed TAL-style analysis for the practical CATE-targeting strategy, Causal-EPIG-$\tau$-S. The goal here is intentionally more modest than in Thm. 4.3. We do *not* claim a full analogue of the global convergence theorem for all MM strategies, since the strong submodularity assumptions required for the global proof need not hold in the general transductive setting. Instead, we show that Causal-EPIG-$\tau$-S can be viewed as an MM-ITL rule on an induced finite-dimensional Gaussian TAL instance whose target variables are scalar CATE predictions.

**Induced Gaussian TAL instance.** Let the target population be $\boldsymbol{X}_{\mathrm{tar}} = \{\boldsymbol{x}_1^*, \ldots, \boldsymbol{x}_m^*\}$. For each target covariate $\boldsymbol{x}^* \in \boldsymbol{X}_{\mathrm{tar}}$, define the scalar CATE target

$$z_{\boldsymbol{x}^*} \coloneqq \tau(\boldsymbol{x}^*) = f((\boldsymbol{x}^*, 1)) - f((\boldsymbol{x}^*, 0)). \tag{39}$$

Introduce an induced finite Gaussian family $g$ over the union of target indices and sample indices by setting $g(a_{\boldsymbol{x}^*}) = z_{\boldsymbol{x}^*}$, $g(s) = f(s)$, $s \in \tilde{\mathcal{D}}_P$. Because each $z_{\boldsymbol{x}^*}$ is a linear functional of the augmented GP $f(\tilde{\boldsymbol{x}})$, the variables $\{g(a_{\boldsymbol{x}^*})\}_{\boldsymbol{x}^* \in \boldsymbol{X}_{\mathrm{tar}}}$ and $\{g(s)\}_{s \in \tilde{\mathcal{D}}_P}$ are jointly Gaussian, with covariance induced by the covariance of $f$. The corresponding target family is

$$\mathcal{A}_\tau \coloneqq \{a_{\boldsymbol{x}^*} : \boldsymbol{x}^* \in \boldsymbol{X}_{\mathrm{tar}}\}, \tag{40}$$

with target variables $g(a_{\boldsymbol{x}^*}) = z_{\boldsymbol{x}^*}$, while the sample space is $\mathcal{S} = \tilde{\mathcal{D}}_P$ with noisy observations $\mathrm{y}_s = g(s) + \varepsilon_s$.

**Identification with MM-ITL.** The mean-marginal Causal-EPIG-$\tau$ utility for a candidate query $\tilde{\boldsymbol{x}} \in \tilde{\mathcal{D}}_P^{(n)}$ is

$$\psi_{\mathrm{MM}}^\tau(\tilde{\boldsymbol{x}} \mid D_n) \coloneqq \frac{1}{|\boldsymbol{X}_{\mathrm{tar}}|} \sum_{\boldsymbol{x}^* \in \boldsymbol{X}_{\mathrm{tar}}} \mathrm{I}(z_{\boldsymbol{x}^*}; \mathrm{y}_{\tilde{\boldsymbol{x}}} \mid D_n). \tag{41}$$

Since $z_{\boldsymbol{x}^*} = g(a_{\boldsymbol{x}^*})$, this is the MM-ITL objective of Hübotter et al. (2024, App. D.3) applied to the induced Gaussian TAL instance $(\mathcal{A}_\tau, \mathcal{S})$. Thus, Causal-EPIG-$\tau$-S corresponds to applying MM-ITL to the target CATE variables $\{\tau(\boldsymbol{x}^*)\}_{\boldsymbol{x}^* \in \boldsymbol{X}_{\mathrm{tar}}}$.

**Additional MM assumption.** Unlike the global ITL analysis, the MM-ITL analysis requires a monotonicity-type condition on the maximal mean-marginal gains. Let

$$\Gamma_n^{\tau,\mathrm{MM}} \coloneqq \max_{\tilde{\boldsymbol{x}} \in \tilde{\mathcal{D}}_P^{(n)}} \psi_{\mathrm{MM}}^\tau(\tilde{\boldsymbol{x}} \mid D_n) \tag{42}$$

denote the maximal MM gain after $n$ active acquisitions over the remaining pool.

**Assumption E.1** (Non-increasing MM maximal gains). The maximal mean-marginal gains satisfy

$$\Gamma_{n+1}^{\tau,\mathrm{MM}} \le \Gamma_n^{\tau,\mathrm{MM}} \qquad \text{for all } n \ge 0. \tag{43}$$

*Remark* E.2. Assumption E.1 is weaker than submodularity but is still not automatic in general transductive settings, where synergistic effects may occur (Hübotter et al., 2024). The result below should therefore be interpreted as a relaxed guarantee for the practical MM strategy, conditional on this monotonicity assumption and the regularity conditions needed for the MM-ITL variance analogue.

**Mean-marginal information capacity and irreducible floor.** Define the average single-target information capacity

$$\bar{\gamma}_n^\tau \coloneqq \frac{1}{|\boldsymbol{X}_{\mathrm{tar}}|} \sum_{\boldsymbol{x}^* \in \boldsymbol{X}_{\mathrm{tar}}} \max_{\tilde{B} \subseteq \tilde{\mathcal{D}}_P,\, |\tilde{B}| \le n} \mathrm{I}(z_{\boldsymbol{x}^*}; \mathrm{y}_{\tilde{B}} \mid D_0). \tag{44}$$

Also define the CATE irreducible uncertainty floor

$$\eta_{\mathcal{D}_P, \tau}^2(\boldsymbol{x}^*) \coloneqq \mathbb{V}\mathrm{ar}\big[z_{\boldsymbol{x}^*} \mid D_0, f_{\tilde{\mathcal{D}}_P}\big]. \tag{45}$$

**Proposition E.3** (Relaxed MM-style bound for Causal-EPIG-$\tau$-S). *Under the augmented GP observation model, finite target and pool spaces, Assumption E.1, and the regularity conditions required for the MM-ITL variance analogue of the TAL convergence theorem, the greedy Causal-EPIG-$\tau$-S rule is covered by the MM-ITL variance analogue applied to the induced Gaussian TAL instance above. Consequently, there exists a constant $C_{\mathrm{MM}} > 0$ such that, for all $1 \le n \le |\tilde{\mathcal{D}}_P|$,*

$$\frac{1}{|\boldsymbol{X}_{\mathrm{tar}}|} \sum_{\boldsymbol{x}^* \in \boldsymbol{X}_{\mathrm{tar}}} \mathbb{V}\mathrm{ar}[\tau(\boldsymbol{x}^*) \mid D_n] \le \frac{1}{|\boldsymbol{X}_{\mathrm{tar}}|} \sum_{\boldsymbol{x}^* \in \boldsymbol{X}_{\mathrm{tar}}} \eta_{\mathcal{D}_P, \tau}^2(\boldsymbol{x}^*) + C_{\mathrm{MM}} \frac{\bar{\gamma}_n^\tau}{\sqrt{n}}. \tag{46}$$

*Proof.* After conditioning on the warm-start set $D_0$, the induced variables $g(a_{\boldsymbol{x}^*}) = z_{\boldsymbol{x}^*}$ and $g(s) = f(s)$ form a finite-dimensional Gaussian TAL problem. The greedy Causal-EPIG-$\tau$-S rule is the corresponding MM-ITL greedy rule, since it maximizes $\frac{1}{|\boldsymbol{X}_{\text{tar}}|} \sum_{\boldsymbol{x}^* \in \boldsymbol{X}_{\text{tar}}} \mathrm{I}(z_{\boldsymbol{x}^*}; \mathrm{y}_{\tilde{\boldsymbol{x}}} \mid D_n)$ over the remaining pool $\tilde{\mathcal{D}}_P^{(n)}$. Applying the MM-ITL variance analogue discussed in Hübotter et al. (2024, App. D.3) to this induced Gaussian instance, under Assumption E.1 and the stated regularity conditions, gives the average posterior-variance bound. $\square$

**Interpretation.** Prop. E.3 is weaker than the global result in Thm. 4.3: it controls marginal CATE uncertanties averaged over the target set, rather than the joint uncertainty of the full target vector. It should be viewed as a relaxed TAL-style support for the scalable CATE-targeting objective used in our experiments, not as a universal convergence theorem for all mean-marginal Causal-EPIG variants.

**Relation to the $\mu$-based mean-marginal strategy.** A similar mean-marginal view applies to the PO-based strategy. For each target covariate $\boldsymbol{x}^*$, define the vector target $\boldsymbol{u}_{\boldsymbol{x}^*} \coloneqq \big(f((\boldsymbol{x}^*, 0)), f((\boldsymbol{x}^*, 1))\big)$. Causal-EPIG-$\mu$-S uses the mean-marginal information gain $\frac{1}{|\boldsymbol{X}_{\text{tar}}|} \sum_{\boldsymbol{x}^* \in \boldsymbol{X}_{\text{tar}}} \mathrm{I}(\boldsymbol{u}_{\boldsymbol{x}^*}; \mathrm{y}_{\tilde{\boldsymbol{x}}} \mid D_n)$, which is the vector-target analogue of MM-ITL. A formal variance bound for this vector-target MM objective would require an analogous vector-valued MM analysis. Nevertheless, marginal PO uncertainty provides a sufficient route to controlling CATE uncertainty. Since $\tau(\boldsymbol{x}^*) = f((\boldsymbol{x}^*, 1)) - f((\boldsymbol{x}^*, 0))$, we have the standard bound

$$\mathbb{V}\mathrm{ar}[\tau(\boldsymbol{x}^*) \mid D_n] \le 2\mathbb{V}\mathrm{ar}[f((\boldsymbol{x}^*, 1)) \mid D_n] + 2\mathbb{V}\mathrm{ar}[f((\boldsymbol{x}^*, 0)) \mid D_n]. \tag{47}$$

Thus, controlling marginal PO uncertainty also provides a valid, though potentially loose, upper bound on marginal CATE uncertainty. A tighter analysis would require explicit control of the cross-arm posterior covariance term, which we leave for future work.

### E.6. Implications of the LMC Kernel Structure

The convergence rate in Thm. 4.3 is governed by the information capacity $\gamma_n$ of the augmented transductive problem. This quantity depends on the joint mutual information $\mathrm{I}(f_{\tilde{\boldsymbol{X}}_{\text{tar}}}; \mathrm{y}_{\tilde{B}} \mid D_0)$ and is therefore determined by the augmented kernel $\tilde{k}$. Under the LMC construction,

$$\tilde{k}((\boldsymbol{x}, t), (\boldsymbol{x}', t')) = \sum_{q=1}^{Q} (B_q)_{t,t'} k_q(\boldsymbol{x}, \boldsymbol{x}'), \tag{48}$$

the off-diagonal entries of the coregionalization matrices $B_q$ encode shared structure between the two potential-outcome surfaces. This shared structure influences the information capacity. If the two outcome surfaces are modeled independently, this corresponds to diagonal $B_q$ matrices and the capacity reflects the complexity of learning two separate functions. When the off-diagonal entries are nonzero, information can transfer across treatment arms through the multitask kernel, potentially reducing the effective complexity of the joint prediction problem. Since the convergence rate depends on $\gamma_n$, this shared structure can affect the magnitude of the bound. We interpret this as a structural advantage of causally aware multitask modeling, rather than a strict comparative guarantee: whether the capacity is substantially reduced depends on the kernel, the target set, and the pool geometry.

### E.7. Connections to Other Causal Strategies

**1. Connection to mean-marginal strategies.** The analysis in App. E.1–E.3 provides a finite-budget convergence characterization for the global, joint PO-based strategy Causal-EPIG-$\mu$-G. This strategy is theoretically natural because it aligns exactly with the Global ITL objective over the augmented target potential outcomes. However, as discussed in App. D.2, the global strategy can be computationally expensive. The practical Causal-EPIG utilities used in our experiments are mean-marginal objectives, which average pointwise information gains over $\boldsymbol{X}_{\text{tar}}$. Prop. E.3 provides a relaxed TAL-style guarantee for the CATE-targeted mean-marginal strategy, while the PO-based mean-marginal strategy admits an analogous marginal-uncertainty interpretation.

**2. Relationship between PO-based and CATE-based targets.** The relationship between the PO-based objective Causal-EPIG-$\mu$ and the CATE-based objective Causal-EPIG-$\tau$ follows from the data processing inequality. For a fixed target covariate $\boldsymbol{x}^*$, let $\boldsymbol{u}_{\boldsymbol{x}^*} = \big(f(\boldsymbol{x}^*, 0), f(\boldsymbol{x}^*, 1)\big)$ and $\tau(\boldsymbol{x}^*) = f(\boldsymbol{x}^*, 1) - f(\boldsymbol{x}^*, 0)$. Since $\tau(\boldsymbol{x}^*)$ is a deterministic function of $\boldsymbol{u}_{\boldsymbol{x}^*}$, for any query response $\mathrm{y}_{\tilde{\boldsymbol{x}}}$ and conditioning information $D_T$,

$$\mathrm{I}(\mathrm{y}_{\tilde{\boldsymbol{x}}}; \boldsymbol{u}_{\boldsymbol{x}^*} \mid D_T) \ge \mathrm{I}(\mathrm{y}_{\tilde{\boldsymbol{x}}}; \tau(\boldsymbol{x}^*) \mid D_T). \tag{49}$$

Thus, the PO-based target is more comprehensive in the sense that it retains the full joint potential-outcome information, while the CATE-based target compresses this information to the differenced estimand. This does not imply that one acquisition rule is uniformly better than the other: a query can be highly informative about prognostic or arm-specific outcome variation while being less informative about the treatment-effect difference. Conversely, when the effect-relevant signal is concentrated in the difference, directly targeting $\tau(\boldsymbol{x}^*)$ can be more sample-efficient. This is the formal basis for the comprehensiveness–focus trade-off discussed in the main paper.

### E.8. Proof of Prop. 4.1

*Proof.* Here $\epsilon_{\text{PEHE}}^{\mathcal{M}}$ denotes the squared PEHE model risk; the square-root PEHE used in experiments is not the object of this equivalence. Let $\mathcal{F}_s$ denote the information available after $s$ acquisition steps, and let $\mathcal{F}_{s+1} = \mathcal{F}_s \vee \sigma(y)$ denote the information after querying candidate $(\boldsymbol{x}, t)$ and observing its factual outcome $y$. We use $\mathbb{E}_{s+1}[\cdot] \equiv \mathbb{E}_{y \sim p(y|\boldsymbol{x}, t, \mathcal{F}_s)}[\cdot]$ for the pre-posterior expectation over the unknown outcome. For a target covariate $\boldsymbol{x}^*$, write $Z(\boldsymbol{x}^*) = \tau(\boldsymbol{x}^*)$. Under squared loss, the Bayes estimator after observing $y$ is the posterior mean $\hat{\tau}_{s+1}(\boldsymbol{x}^*) = \mathbb{E}[Z(\boldsymbol{x}^*) \mid \mathcal{F}_{s+1}]$. The model-based posterior risk at $\boldsymbol{x}^*$ is $\mathbb{E}\big[(\hat{\tau}_{s+1}(\boldsymbol{x}^*) - Z(\boldsymbol{x}^*))^2 \mid \mathcal{F}_{s+1}\big] = \mathbb{V}\text{ar}[Z(\boldsymbol{x}^*) \mid \mathcal{F}_{s+1}]$. Therefore, the expected model-based PEHE after acquiring $(\boldsymbol{x}, t)$ is

$$\mathbb{E}_{s+1}\big[\epsilon_{\text{PEHE}}^{\mathcal{M}}(\hat{\tau}_{s+1})\big] = \mathbb{E}_{s+1}\big[\mathbb{E}_{p_{\text{tar}}(\boldsymbol{x}^*)}\big[\mathbb{V}\text{ar}[\tau(\boldsymbol{x}^*) \mid \mathcal{F}_{s+1}]\big]\big] \tag{50}$$

$$= \mathbb{E}_{p_{\text{tar}}(\boldsymbol{x}^*)}\big[\mathbb{E}_{s+1}\big[\mathbb{V}\text{ar}[\tau(\boldsymbol{x}^*) \mid \mathcal{F}_{s+1}]\big]\big], \tag{51}$$

where the second equality follows by Fubini's theorem. Hence, the one-step model-based acquisition objective is

$$\arg\min_{(\boldsymbol{x}, t) \in D_P} \mathbb{E}_{p_{\text{tar}}(\boldsymbol{x}^*)}\big[\mathbb{E}_{s+1}\big[\mathbb{V}\text{ar}[\tau(\boldsymbol{x}^*) \mid \mathcal{F}_{s+1}]\big]\big]. \tag{52}$$

For GP models with fixed hyperparameters and Gaussian likelihood, the posterior covariance after adding a candidate input $(\boldsymbol{x}, t)$ is deterministic given $\mathcal{F}_s$ and the candidate input; it does not depend on the realized value of $y$. Thus the inner expectation over $y$ can be dropped:

$$\arg\min_{(\boldsymbol{x}, t) \in D_P} \mathbb{E}_{p_{\text{tar}}(\boldsymbol{x}^*)}\big[\mathbb{V}\text{ar}[\tau(\boldsymbol{x}^*) \mid \mathcal{F}_{s+1}]\big]. \tag{53}$$

Finally, since $\tau(\boldsymbol{x}^*) = f(\boldsymbol{x}^*, 1) - f(\boldsymbol{x}^*, 0)$, the posterior CATE variance is

$$\mathbb{V}\text{ar}[\tau(\boldsymbol{x}^*) \mid \mathcal{F}_{s+1}] = \mathbb{V}\text{ar}[f(\boldsymbol{x}^*, 1) \mid \mathcal{F}_{s+1}] + \mathbb{V}\text{ar}[f(\boldsymbol{x}^*, 0) \mid \mathcal{F}_{s+1}] \tag{54}$$

$$- 2\text{Cov}[f(\boldsymbol{x}^*, 1), f(\boldsymbol{x}^*, 0) \mid \mathcal{F}_{s+1}]. \tag{55}$$

Therefore, the optimal one-step model-based acquisition criterion is to minimize the integrated posterior CATE variance, which depends on both marginal potential-outcome uncertainties and their posterior covariance. $\square$

#### E.8.1. CONNECTION TO THE CAUSAL-EPIG FRAMEWORK

Prop. 4.1 shows that, under the GP and model-based squared-error assumptions, the natural one-step objective is to minimize integrated posterior CATE variance. Causal-EPIG provides information-theoretic acquisition rules that target the causal quantities determining this variance.

**Causal-EPIG-$\mu$ (Comprehensive Strategy).** This strategy maximizes

$$\arg\max_{(\boldsymbol{x}, t) \in D_P} \mathbb{E}_{p_{\text{tar}}(\boldsymbol{x}^*)}\Big[\text{I}\big(y; (f(\boldsymbol{x}^*, 0), f(\boldsymbol{x}^*, 1)) \mid (\boldsymbol{x}, t), D_T\big)\Big]. \tag{56}$$

For Gaussian posteriors, this is equivalent to maximizing the expected reduction in the joint entropy of the potential-outcome pair. Since the CATE variance depends on the marginal variances of $f(\boldsymbol{x}^*, 0)$ and $f(\boldsymbol{x}^*, 1)$ as well as their covariance, this objective targets the joint posterior structure from which $\mathbb{V}\text{ar}[\tau(\boldsymbol{x}^*)]$ is constructed.

**Causal-EPIG-$\tau$ (Focused Strategy).** This strategy maximizes information gain about the estimand itself:

$$\arg\max_{(\boldsymbol{x}, t) \in D_P} \mathbb{E}_{p_{\text{tar}}(\boldsymbol{x}^*)}\Big[\text{I}\big(y; \tau(\boldsymbol{x}^*) \mid (\boldsymbol{x}, t), D_T\big)\Big]. \tag{57}$$

Equivalently, it minimizes the expected posterior entropy of $\tau(\boldsymbol{x}^*)$. For Gaussian posteriors, $\mathrm{H}(\tau(\boldsymbol{x}^*) \mid D) = \frac{1}{2}\log(2\pi e\,\mathbb{V}\mathrm{ar}[\tau(\boldsymbol{x}^*) \mid D])$. Under the GP assumption, the posterior variance after querying $(\boldsymbol{x}, t)$ is deterministic with respect to the future outcome $y$, so the Causal-EPIG-$\tau$ objective is equivalent to

$$\underset{(\boldsymbol{x},t)\in D_P}{\arg\min}\, \mathbb{E}_{p_{\mathrm{tar}}(\boldsymbol{x}^*)}\left[\log \mathbb{V}\mathrm{ar}[\tau(\boldsymbol{x}^*) \mid \mathcal{F}_{s+1}]\right]. \tag{58}$$

This is not identical to minimizing the integrated variance in Prop. 4.1; it is the corresponding log-variance, entropy-based uncertainty criterion. Thus, Causal-EPIG-$\tau$ is a principled information-theoretic proxy that directly targets posterior uncertainty in the final CATE estimand. Together, Causal-EPIG-$\mu$ and Causal-EPIG-$\tau$ provide two aligned but distinct acquisition targets: the former preserves the full joint potential-outcome structure, while the latter focuses directly on the differenced causal estimand.

### E.9. Relationship between Causal-EPIG and Parameter-Based Acquisition

**Proposition E.4.** *Assume that, for any target covariate $\boldsymbol{x}^*$, the target causal prediction quantity is a measurable function of the corresponding model parameters. Then the parameter-oriented information utilities upper-bound their prediction-oriented counterparts:*

$$(a) \quad \text{Causal-BALD} - \mu(\boldsymbol{x}, t) \geq \text{Causal-EPIG} - \mu(\boldsymbol{x}, t), \tag{59}$$

$$(b) \quad \text{Causal-EIG}(\boldsymbol{x}, t) \geq \text{Causal-EPIG} - \tau(\boldsymbol{x}, t). \tag{60}$$

*For models without a distinct CATE-parameter block, part (b) should be read with $\theta_\tau$ replaced by any parameter or latent representation that determines the CATE process.*

*Proof.* Let $D_T' = D_T \cup \{(\boldsymbol{x}, t)\}$ denote the acquisition context containing the candidate covariates and treatment, before observing the random outcome $y$. The relevant utilities can be written as

$$\text{Causal-BALD} - \mu(\boldsymbol{x}, t) \coloneqq \mathrm{I}(y; \theta \mid D_T'),$$

$$\text{Causal-EPIG} - \mu(\boldsymbol{x}, t) \coloneqq \mathbb{E}_{p_{\mathrm{tar}}(\boldsymbol{x}^*)}\left[\mathrm{I}\big(y; (f(\boldsymbol{x}^*, 0), f(\boldsymbol{x}^*, 1)) \mid \boldsymbol{x}^*, D_T'\big)\right],$$

$$\text{Causal-EIG}(\boldsymbol{x}, t) \coloneqq \mathrm{I}(y; \theta_\tau \mid D_T'),$$

$$\text{Causal-EPIG} - \tau(\boldsymbol{x}, t) \coloneqq \mathbb{E}_{p_{\mathrm{tar}}(\boldsymbol{x}^*)}\left[\mathrm{I}\big(y; \tau(\boldsymbol{x}^*) \mid \boldsymbol{x}^*, D_T'\big)\right].$$

For part (a), conditional on $(\boldsymbol{x}^*, D_T')$, the target latent potential-outcome pair is downstream of the model parameters $\theta$. Hence $y \longrightarrow \theta \longrightarrow (f(\boldsymbol{x}^*, 0), f(\boldsymbol{x}^*, 1))$ is a conditional Markov chain. By the conditional data processing inequality,

$$\mathrm{I}(y; \theta \mid \boldsymbol{x}^*, D_T') \geq \mathrm{I}\big(y; (f(\boldsymbol{x}^*, 0), f(\boldsymbol{x}^*, 1)) \mid \boldsymbol{x}^*, D_T'\big). \tag{61}$$

Since $\boldsymbol{x}^*$ is sampled independently of $(y, \theta)$ conditional on $D_T'$,

$$\mathrm{I}(y; \theta \mid \boldsymbol{x}^*, D_T') = \mathrm{I}(y; \theta \mid D_T') = \text{Causal-BALD} - \mu(\boldsymbol{x}, t). \tag{62}$$

Taking expectation over $\boldsymbol{x}^* \sim p_{\mathrm{tar}}$ gives Causal-BALD $- \mu(\boldsymbol{x}, t) \geq$ Causal-EPIG $- \mu(\boldsymbol{x}, t)$.

For part (b), conditional on $(\boldsymbol{x}^*, D_T')$, the CATE prediction $\tau(\boldsymbol{x}^*)$ is downstream of the CATE parameters $\theta_\tau$, or the corresponding CATE-determining latent representation. This gives the conditional Markov chain $y \longrightarrow \theta_\tau \longrightarrow \tau(\boldsymbol{x}^*)$. Thus,

$$\mathrm{I}(y; \theta_\tau \mid \boldsymbol{x}^*, D_T') \geq \mathrm{I}\big(y; \tau(\boldsymbol{x}^*) \mid \boldsymbol{x}^*, D_T'\big). \tag{63}$$

Again, because $\boldsymbol{x}^*$ is independent of $(y, \theta_\tau)$ conditional on $D_T'$,

$$\mathrm{I}(y; \theta_\tau \mid \boldsymbol{x}^*, D_T') = \mathrm{I}(y; \theta_\tau \mid D_T') = \text{Causal-EIG}(\boldsymbol{x}, t). \tag{64}$$

Taking expectation over $\boldsymbol{x}^* \sim p_{\mathrm{tar}}$ gives Causal-EIG$(\boldsymbol{x}, t) \geq$ Causal-EPIG $- \tau(\boldsymbol{x}, t)$. $\square$

Equality in the inequalities above holds when the prediction quantity retains all information about the corresponding parameters that is relevant to the candidate outcome. For example, part (b) is tight when $y \perp\!\!\!\perp \theta_\tau \mid \tau(\boldsymbol{x}^*), \boldsymbol{x}^*, D_T'$. Otherwise, the parameter-focused objective can contain additional information about parameter directions that do not affect the target causal prediction. This is why the inequalities should not be interpreted as a sample-efficiency guarantee for parameter-based acquisition: they show that parameter-based targets are more comprehensive in an information-theoretic sense, whereas Causal-EPIG deliberately focuses the acquisition target on the causal predictions of interest.

# F. Further Experimental Results

This section provides supplementary experimental results. We first report full test-set convergence curves for all benchmark datasets and base CATE estimators. We then provide robustness ablations on warm-start size, unlabeled pool size, acquisition batch size, and the Deep Variational GP estimator used in the original Causal-BALD study.

## F.1. Benchmark Convergence Curves

**CausalBALD synthetic dataset.** Fig. 8 and Fig. 9 show test-set $\sqrt{\text{PEHE}}$ on the CausalBALD synthetic benchmark under regular and target-shift settings.

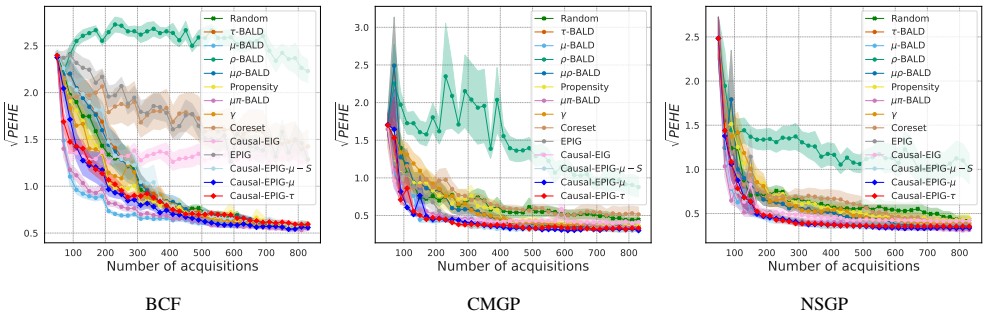

*Figure 8.* Test-set performance on the CausalBALD synthetic dataset under the regular setup. Each panel reports $\sqrt{\text{PEHE}}$ over the acquisition trajectory for a different base CATE estimator.

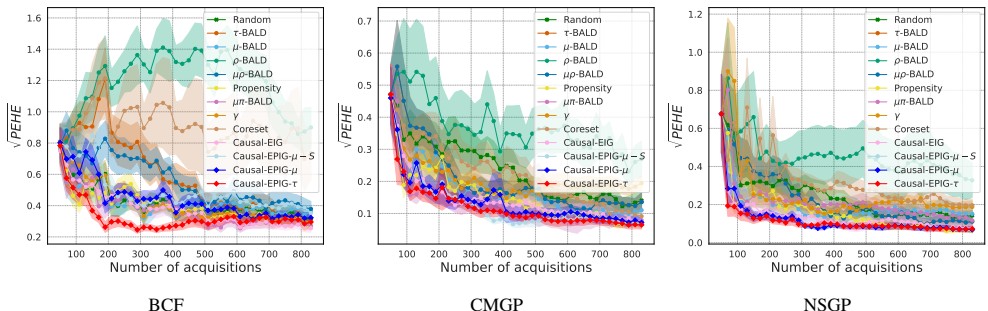

*Figure 9.* Test-set performance on the CausalBALD synthetic dataset under target distribution shift. Each panel reports $\sqrt{\text{PEHE}}$ over the acquisition trajectory for a different base CATE estimator.

**Hahn synthetic dataset.** Fig. 10–13 report the corresponding test-set convergence curves for the Hahn linear and nonlinear settings, with and without target distribution shift.

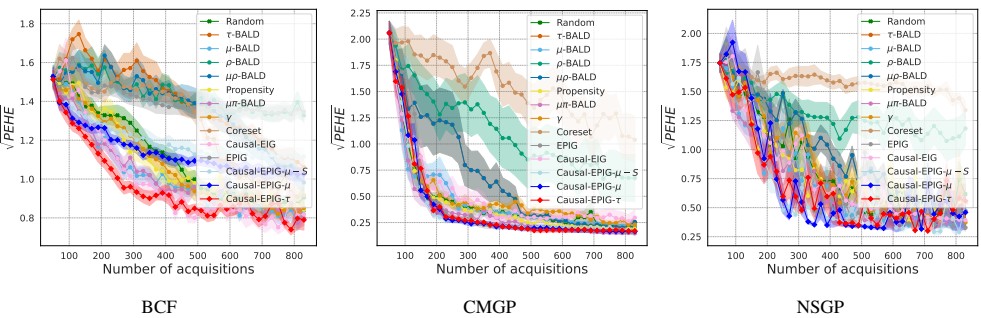

*Figure 10.* Test-set performance on the Hahn linear synthetic dataset under the regular setup.

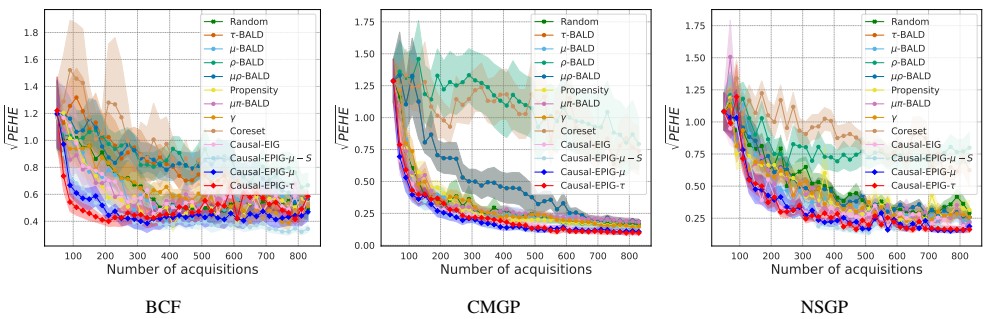

*Figure 11.* Test-set performance on the Hahn linear synthetic dataset under target distribution shift.

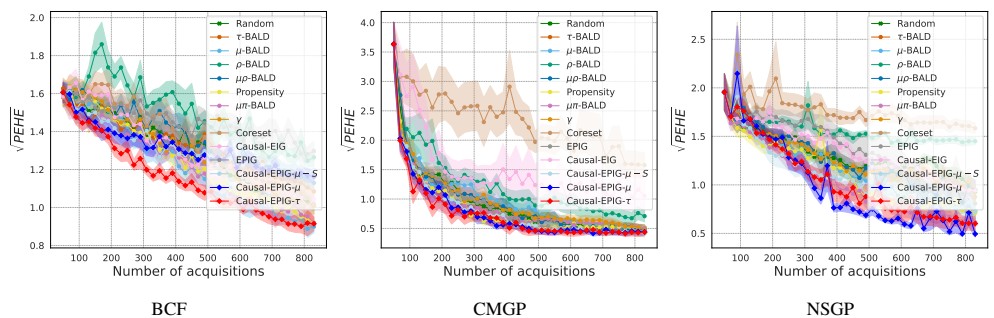

*Figure 12.* Test-set performance on the Hahn nonlinear synthetic dataset under the regular setup.

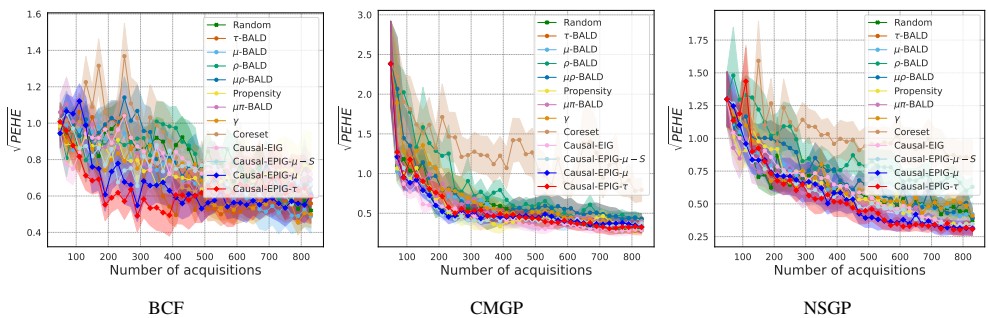

*Figure 13.* Test-set performance on the Hahn nonlinear synthetic dataset under target distribution shift.

**Semi-synthetic datasets.** Fig. 14–16 report test-set convergence curves for IHDP and ACTG-175.

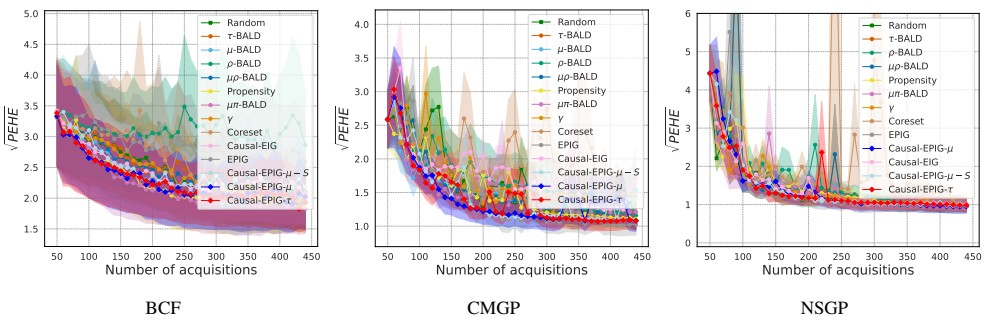

*Figure 14.* Test-set performance on the IHDP semi-synthetic dataset under the regular setup.

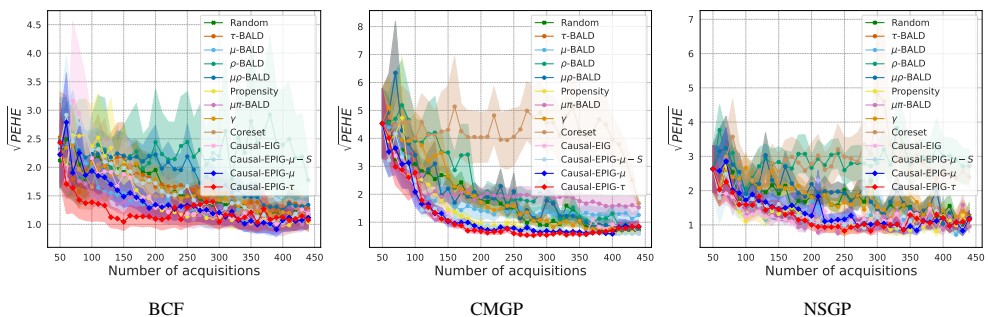

*Figure 15.* Test-set performance on the IHDP semi-synthetic dataset under target distribution shift.

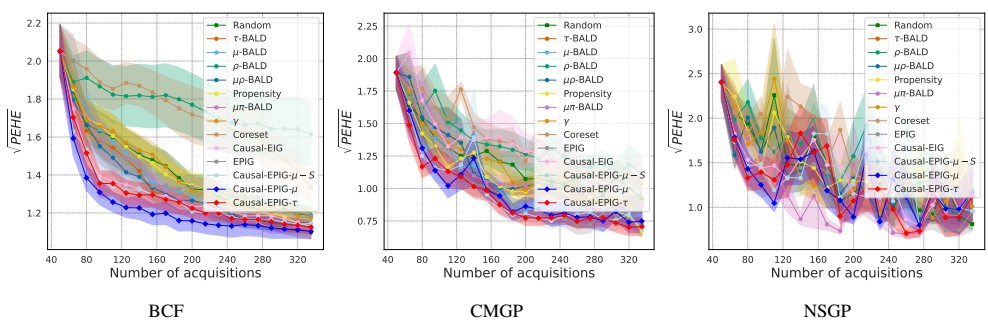

*Figure 16.* Test-set performance on the ACTG-175 semi-synthetic dataset under the standard setup.

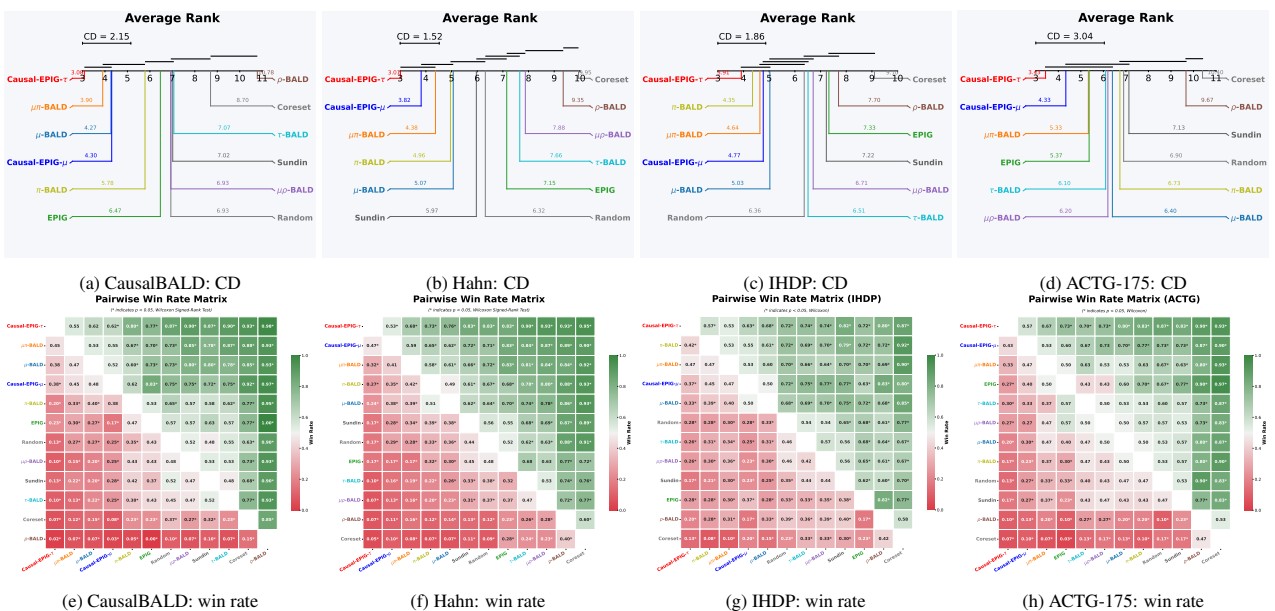

*Figure 17.* Rank-based comparison of acquisition functions across benchmark groups. The top row shows critical difference (CD) diagrams, and the bottom row shows pairwise win-rate matrices with statistical significance. Columns correspond to the CausalBALD synthetic benchmark, Hahn synthetic benchmarks, IHDP, and ACTG-175.

## F.2. Robustness Ablations

We conduct ablations on the Hahn linear dataset with the BCF estimator to assess sensitivity to active-learning hyperparameters. Across these studies, Causal-EPIG maintains strong relative performance, suggesting that its gains are not driven by a single warm-start size, pool size, or batch size.

**Warm-start size.** Fig. 18 varies the number of initial random samples from 50 to 300. Larger warm starts reduce the initial PEHE for all methods, but the relative ranking remains stable.

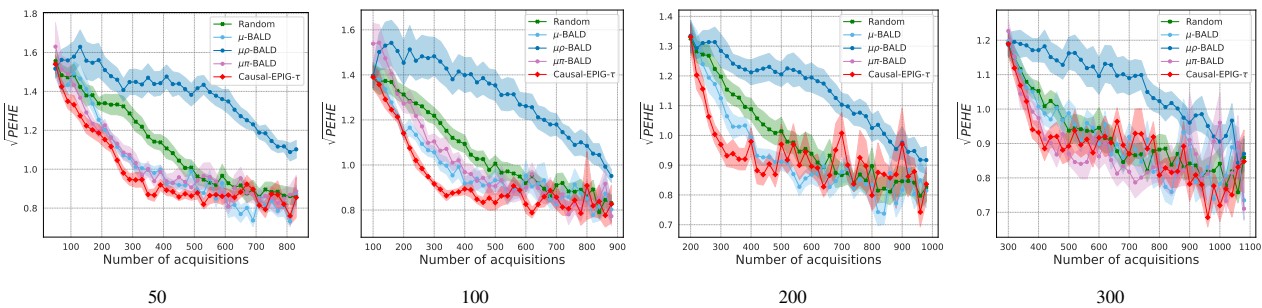

*Figure 18.* Sensitivity to warm-start size on the Hahn linear dataset with BCF. Panels correspond to 50, 100, 200, and 300 initial random samples.

**Pool size.** Fig. 19 varies the initial unlabeled pool size from 1000 to 2500. Larger pools give active methods a broader candidate set, but the relative performance trend remains similar.

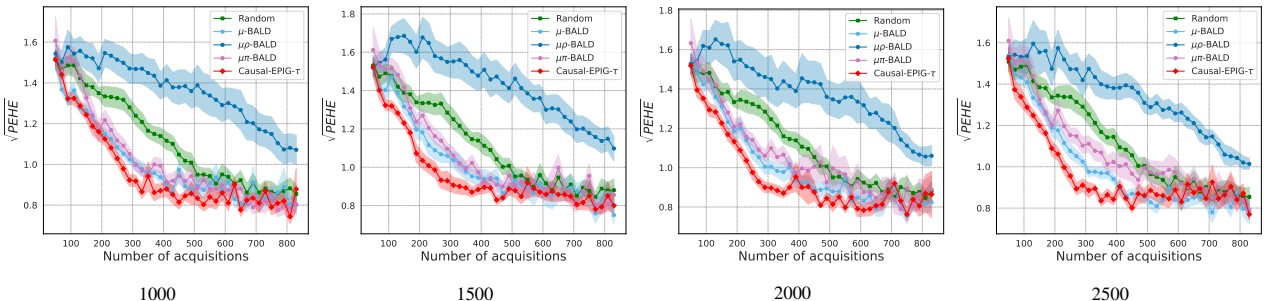

*Figure 19.* Sensitivity to unlabeled pool size on the Hahn linear dataset with BCF. Panels correspond to $|D_P| = 1000, 1500, 2000, 2500$.

**Batch size.** Fig. 20 varies the acquisition batch size from 5 to 40. Smaller batches allow more frequent model updates, while larger batches reduce retraining cost; Causal-EPIG remains competitive across this range.

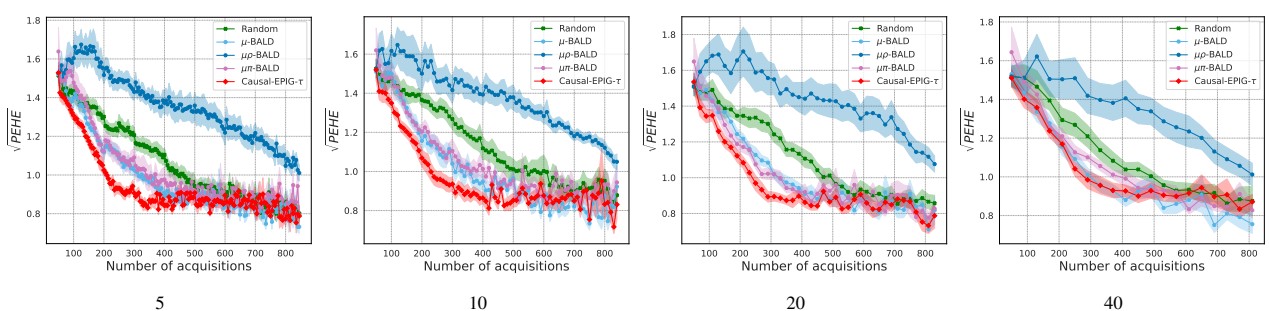

*Figure 20.* Sensitivity to acquisition batch size on the Hahn linear dataset with BCF. Panels correspond to $n_b = 5, 10, 20, 40$.

**Deep Variational GP estimator.** To compare with the estimator used in the original Causal-BALD study (Jesson et al., 2021), we also evaluate the Deep Variational GP estimator. Fig. 21 shows that Causal-EPIG remains competitive across these settings, indicating that its behavior is not tied only to BCF or exact GP estimators.

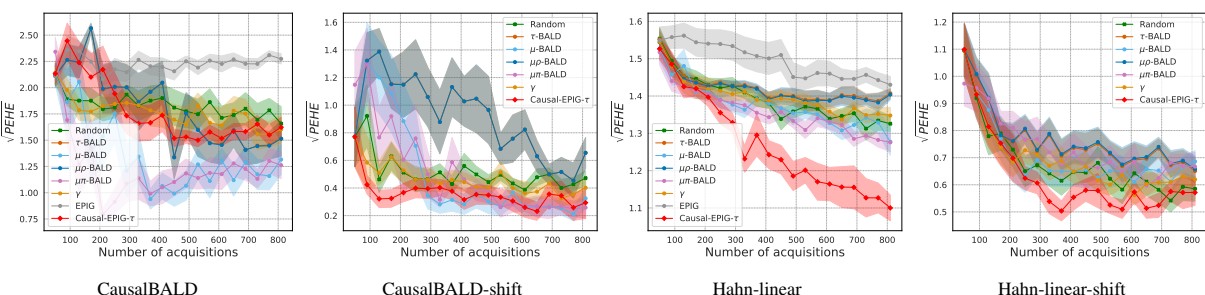

*Figure 21.* Performance with the Deep Variational GP estimator used in the original Causal-BALD study. Panels show CausalBALD, CausalBALD with shift, Hahn linear, and Hahn linear with shift.

### F.3. Targeted Synthetic Study for Causal-EPIG-$\mu$ vs. Causal-EPIG-$\tau$

We include a targeted synthetic study to isolate when the PO-based and CATE-based Causal-EPIG objectives are preferred. The covariates are partitioned into prognostic, predictive, and noise blocks. We vary the predictive share of the active informative block, $\rho = \frac{k_{\text{pred}}}{k_{\text{prog}}+k_{\text{pred}}}$, while keeping the total active informative dimension fixed. Thus, changing $\rho$ changes the organization of the informative structure rather than simply changing the number of active variables. We compare two treatment-effect formation mechanisms. In the *surface-split* regime, the effect is induced by the difference of two arm-specific outcome surfaces. Schematically, $\mu_0(x) = g(p) + c(z_C)$ and $\mu_1(x) = g(p) + t(z_T)$, so that $\tau(x) = t(z_T) - c(z_C)$. Here, the treatment-effect signal is distributed across the two potential-outcome surfaces, so learning the joint PO structure remains useful. In the *cancellation* regime, a substantial part of the full-outcome structure is informative for potential-outcome prediction but cancels, or becomes weakly relevant, in the treatment-effect difference. A schematic form is $\mu_0(x) = g(p) + s(x) - \frac{1}{2}r(x)$ and $\mu_1(x) = g(p) + s(x) + \frac{1}{2}r(x)$, so that the differenced target is dominated by the residual effect-bearing signal $r(x)$. In this regime, directly targeting the CATE can be more efficient. The acquisition trajectories are shown in Fig. 5. The surface-split setting favors Causal-EPIG-$\mu$, while the cancellation settings increasingly favor Causal-EPIG-$\tau$ as $\rho$ grows. These results support the interpretation that the relevant factor is not merely the number of predictive or prognostic variables, but how the treatment-effect signal is organized relative to the full outcome structure.

