# OpenReview forum: "Causal-EPIG: Causally Aligned Active CATE Estimation"
_ICML.cc/2026/Conference — ICML 2026 regular_

### Official Review · Reviewer_jKQR · 2026-03-04

**Soundness:** 4
**Presentation:** 3
**Significance:** 3
**Originality:** 2
**Overall Recommendation:** 4
**Confidence:** 4

**Summary:**

This paper applies the EPIG active learning acquisition function to the CATE estimation setting. The authors argue that in the case of CATE estimation, the predictive object of the acquisition function should be a causal quantity, such as the potential outcomes or the treatment effect itself, which leads the authors to propose Causal-EPIG-$\mu$ and Causal-EPIG-$\tau$. The authors show how these acquisition functions are related to the theoretically optimal strategy, and then empirically demonstrate how they compare to multiple baseline acquisition functions across multiple synthetic and semi-synthetic datasets.

**Compliance With Llm Reviewing Policy:**

Affirmed.

**Final Justification:**

The authors adapt the EPIG acquisition strategy to the treatment effect estimation setting. While this is a pretty straightforward application of the existing method to a new setting, the authors do a thorough evaluation on several datasets, and complement it with theoretical analysis. The new results added during the rebuttal help to further illuminate the differences between Causal-EPIG-$\mu$ and Causal-EPIG-$\tau$ as well as between Causal-EPIG and parameter based approaches such as Causal-BALD. Overall, I think that while straightforward, this is a solid work, filling in a gap in the causal inference literature.

**Key Questions For Authors:**

1. Could the authors provide more details about the estimation procedure for the Causal-EPIG objectives? It seems to me that the estimation relies on samples from the models for each potential outcome function $f(x, t | \theta)$ and for the CATE function $\tau(x|\theta)$, although this is not fully clear. I think for the purpose of clarity it would be beneficial to include a detailed algorithmic procedure for calculating the Causal-EPIG acquisition functions using these models.
2. What is $D_T’$?

**Limitations:**

I think the limitations of the Causal-EPIG method, particularly compared to other acquisition functions, could be described in more details, to give an understanding for when using this acquisition function could lead to sub-optimal performance. In particular, the authors mention that the success relies on the model's calibration, which might be poor in low-data regime (i.e. at the beginning of the acquisition project). It would be interesting to expand on this discussion (and ideally, also see experiments of when this can happen).

**Strengths And Weaknesses:**

## Strengths

This paper is applying the EPIG acquisition strategy to the causal setting, for the purpose of more efficient CATE estimation. I think this is an interesting problem, important for downstream applications when obtaining the outcomes is costly, and constitutes a significant (although straight-forward and quite narrow) gap in the literature.

The authors describe the problem setting and the motivation for using causally-oriented EPIG very clearly. Related works are also (mostly) discussed in detail.

Theoretically, the paper establishes some notion of “optimal” greedy acquisition policy under particular assumptions (Proposition 4.1) and then discuss how their proposed approach approximates this policy. This helps to better justify the use of their proposed acquisition functions.

Empirically, the authors compare against an exhaustive list of acquisition baselines, and run experiments on multiple datasets. The authors also carry out significant pairwise comparisons and statistical analysis to quantify the gains obtained with their method. The authors also consider the batch-based (non-greedy) extension of their acquisition strategy, which I think is a great addition to the paper.

Overall, I really like the soundness of the paper and I see it as its main strength.

## Weaknesses

**Limited analysis:** Given that the idea of applying EPIG in the CATE estimation setting is pretty straightforward, thus limiting the contribution and significance of the paper, I would expect the authors to provide more detailed and comprehensive analysis of when this approach might provide benefits, compared to other approaches. Specifically:

1. When is this attitude better than the parameter-based acquisition presented in Causal-BALD?As per the original EPIG paper, the answer seems to be “when there is a distribution shift between the sampling pool and the target distribution”, however it seems to me that this has not been sufficiently analysed (e.g., through a targeted ablation where increasing the distribution shift is shown to lead to a larger performance gap). Further, for the experiments targeting the distribution shift setting, it is not clear to me how such distribution shift is obtained and how its strength is modulated.
2. When is Causal-EPIG-$\mu$ expected to work better than Causal-EPIG-$\tau$? The authors mention that “the choice between them embodies a context-dependent trade-off”, but it is not clear what factors affect the difference. The authors briefly discuss that the difference between the two is related to the strength of the prognostic and predictive factors in the CATE function. However, again, I do believe that at least the empirical analysis here could be strengthened and much more detailed. See [1] for an example of such analysis carried out in a very related context of active feature acquisition for CATE estimation, where the authors also consider different acquisition functions based on different causal quantities. One possible analysis could look at the performance of the efficiency of two acquisition functions change as we increase the number of predictive and prognostic factors in the data generating process (this shouldn’t be too difficult given that all the datasets considered are synthetic or semi-synthetic).
Overall, I think that strengthening this type of targeted analysis could significantly improve the originality and significance of the paper.

**Clarity could be improved:**
- Currently, a significant chunk of the paper is dedicated to explaining the need for *causally aligned* acquisition functions (significant part of the introduction + section 3). I personally think that these observations are not particularly surprising, and by shortening that part of the paper, the authors could recover some extra space which could be used for improving the clarity of the estimation procedure, which is only discussed very briefly in section 4.2.
- Figure 2 is very difficult to parse as it is not sufficiently labelled. Could the authors make explicit on the plot what each column and row represents? Why does EPIG appear only on selected graphs? Do the authors have any intuition for why the curves are not monotonically decreasing? Additionally, the density of the baselines considered makes it difficult to draw comparisons. While the relative improvement plots (Figure 3) makes this more explicit, it is not clear what the different dots represent in there (is it seeds? for how many acquisitions were these results aggregated?)

**Other:**
- Theorem 4.4 relies on the assumption that the Causal-EPIG-$\mu$-G function is submodular. Have the authors tried checking/proving whether it actually is? It seems like a missing piece in the theoretical analysis. The theoretical result presented in [1] can be a helpful starting point, proving a similar result for the *conditional* mutual information. Further, I think that Theorem 4.4 merits some additional discussion about its implications in this context, as it is not clear to me what the main conclusion from this analysis is.

[1] Piskorz, J., Astorga, N., Berrevoets, J. & van der Schaar, M.. (2025). Active Feature Acquisition for Personalised Treatment Assignment. *Proceedings of The 28th International Conference on Artificial Intelligence and Statistics*, in *Proceedings of Machine Learning Research* 258:4330-4338 Available from https://proceedings.mlr.press/v258/piskorz25a.html.

---

> ### Author Rebuttal · Authors · 2026-03-30
>
> Thank you for your rigorous review for improvement, which we will gladly incorporate. Where your queries overlap with others, we have provided unified answers to conserve space.
>
> **W1** See W.1 of Reviewer DCfA.
>
> **W2** We conducted a targeted synthetic study with systematic variation of: the number of prognostic vs.\ predictive factors, and the treatment-effect formation mechanism, directly following the suggested DGP design. The results reveal two clear regimes:
>
> - **Surface-split (effect as difference of outcome surfaces):** Causal-EPIG-$\mu$ is consistently more efficient, as learning both potential-outcome surfaces directly improves $\tau(x)$ estimation.
> - **Cancellation (predictive structure cancels in $\tau$):** Causal-EPIG-$\tau$ is preferred, as it focuses on the residual treatment-effect signal rather than irrelevant predictive variation.
>
> These regimes provide a clear answer to when each objective is preferable: $\mu$ is favored when the treatment effect is induced by differences in outcome surfaces, whereas $\tau$ is favored when predictive structure cancels and the signal lies in residual effects.
>
> Predictive vs.\ prognostic composition further modulates this behavior: as the number of predictive factors increases, the advantage shifts toward Causal-EPIG-$\tau$ in the cancellation regime. Quantitatively, under cancellation with low predictive share, $\mu$ improves PEHE@200 from $1.04$ to $0.83$, while under stronger predictive structure the advantage shifts to $\tau$. These results provide a concrete, empirically validated, mechanism-driven explanation of when each objective is preferable.
>
>
> **W3** We will streamline the introduction and Sec. 3 to recover space for improving Sec. 4.2. In particular, we will provide a clearer algorithmic description of Causal-EPIG, detailing both GP-based and sample-based implementations, including how predictive samples are used to estimate the mutual information quantities in practice.
>
> **W4** Fig. 2 is currently under-labeled. We will revise it to explicitly indicate: rows = base learners (BCF, CMGP, NSGP), columns = dataset/setting (CausalBALD vs. Hahn; regular vs. distribution shift). EPIG appears only in selected plots due to incomplete reporting of distribution-shift runs; we have re-run and included these results, with conclusions unchanged.
>
> We will also improve clarity by reducing visual density (clearer legends, highlighting key methods). In Fig. 3, each dot corresponds to one acquisition step; results are aggregated over seeds and models to form mean PEHE curves, and box plots summarize per-step relative improvements over the Random baseline.
>
> **W5** See Q2 of Reviewer Wyev.
>
> **Q1** We will describe the estimation procedure more explicitly and include a clear algorithmic formulation of Causal-EPIG. For sample-based models, we use posterior samples to construct predictive quantities for both $f(x,t\mid\theta)$ and $\tau(x\mid\theta)$, from which the required covariance structures are estimated. Concretely, for each candidate $x$: (1) draw posterior samples of the relevant predictive quantities (either $f(x,t)$ or $\tau(x)$, depending on the variant); (2) construct empirical covariance matrices over the target set $X_{\mathrm{tar}}$; (3) compute mutual information using the Gaussian MI formula based on these covariances.
>
> For GP-based models, the same quantities are obtained in closed form from the predictive covariance (Eq.~6), without sampling. Thus, the only difference is how predictive uncertainty is obtained (closed-form vs.\ sample-based), while the acquisition computation itself remains identical.
>
> **Q2** $D'_T$ denotes the updated training dataset after the acquisition step.
>
> **L1** We conducted additional experiments to characterize when Causal-EPIG becomes sub-optimal, focusing on posterior calibration:
>
> - **Not uniformly fragile in low-data regimes.** Even under severe miscalibration (poor coverage, high NLL), Causal-EPIG-$\mu$ still outperforms marginal baselines in acquisition quality (e.g., Top-1 utility $0.27$ vs. $-0.06$, with substantially lower regret), so low-data alone does not imply failure.
> - **Failure is driven by joint miscalibration.** A controlled covariance ablation (removing cross-covariances while keeping marginals fixed) leads to sharp degradation (Top-1 utility $\approx 0$, much higher regret), while $\mu$-BALD remains unchanged, isolating reliance on joint posterior structure.
> - **Severe misspecification affects all methods.** In adversarial regimes, both EPIG and BALD exhibit poor ranking (negative correlation, high regret), indicating the issue is model misspecification rather than low data alone.
>
> Causal-EPIG is therefore most sensitive to failures in modeling joint dependence, and can be sub-optimal when this structure is poorly estimated. When joint uncertainty is reliable, it remains effective even in low-data regimes; when it is not, marginal methods can be more robust.

---

> > ### Author Rebuttal · Reviewer_jKQR · 2026-04-03
> >
> > Thank you for your extensive rebuttal -- I do believe that the provided analysis, discussion and experimental results have allowed to significantly improve the quality of this paper. I am willing to increase my score to 4, but would first kindly ask the authors to provide some additional clarifications and details in the additional space available. Namely
> > 1. Regarding W1, there are a few statements in the response to the reviewer DCfA that I find slightly confusing, such as "the target covariates define where the CATE must be accurate" or "Prediction-based acquisition can therefore be advantageous even without marginal shift, as it directly targets uncertainty in the target CATE rather than global parameter uncertainty" --> if the model is trained to predict the CATE, shouldn't parameter uncertainty reflect the CATE uncertainty? Could the authors please further clarify their claims there?
> > 2. Regarding the new results provided for W2 and L1 above, could the authors provide the relevant figures/tables using anonymised links, to allow me to better investigate the described results?
> >
> > Thank you for the great work on the rebuttal!

---

> > > ### Author Response · Authors · 2026-04-07
> > >
> > > Happy holidays, and apologies for the delayed reply. We thank the reviewer for the positive feedback and the opportunity to further clarify these points.
> > >
> > > Fig Link: https://anonymous.4open.science/r/ICML26-rebuttal-B3F0
> > >
> > > **[Q1]** (1) This means our objective is CATE accuracy on the *target set* $X_{tar}$, not the full covariate space.
> > >
> > > (2) Not all estimator parameters are directly relevant to the CATE quantity. This is exactly the motivation behind Causal-EIG[1], which improves over global parameter uncertainty by targeting only the *causal parameters*. Our point goes one step further: we directly target uncertainty in the *causal estimand* itself over the target distribution, rather than uncertainty in parameters. These two notions coincide only in restricted cases. These coincide only in restricted cases; thus, even without shift, target-estimand and parameter uncertainty need not align on $X_{tar}$.
> > >
> > > [1] Is Merging Worth It? Securely Evaluating the Information Gain for Causal Dataset Acquisition, AISTATS 2025.
> > >
> > > **[W2]** (Fig. 1) The key distinction is target-level. CEPIG-$\mu$ measures IG about the joint PO pair $(y(0),y(1))$, retaining covariance structure (unlike $R_{PO}$ in AFA), CEPIG-$\tau$ targets the differenced quantity $y(1)-y(0)$ (analogous in spirit to $R_{\tau}$). Thus, the issue is not simply how many prognostic or predictive variables are present, but how the treatment-effect (TE) signal is organized relative to the full outcome structure. *Both acq functions depend on joint PO structure; the difference is that $\tau$-targeting compresses the target to the differenced signal, $\mu$-targeting preserves the full joint outcome structure. This also explains why, in our paper, the gap is often modest*. We therefore deliberately construct two synthetic regimes to isolate when each target is preferred.
> > >
> > > In the *surface-split* regime, TE is induced by the difference of two arm-specific outcome surfaces,
> > > $\mu_0(x)=g(p)+c(z_C),\ \mu_1(x)=g(p)+t(z_T)$, so that $\tau(x)=t(z_T)-c(z_C)$. Here, the effect is distributed across the two outcome surfaces rather than concentrated in a separate residual channel. Thus, improving TE estimation still mainly requires learning the full PO pair well, favoring CEPIG-$\mu$.
> > >
> > > In the *cancellation* regime, we construct the outcomes so that a substantial part of the full-outcome structure is informative for $y(0),y(1)$ but cancels, or becomes much less relevant, in the TE difference. A useful schematic is $\mu_0(x)=g(p)+s(x)-\tfrac12 r(x),\ \mu_1(x)=g(p)+s(x)+\tfrac12 r(x)$,
> > > so that the differenced target is dominated by the residual effect-bearing signal $r(x)$. Here, $s(x)$ contributes to outcome complexity but is weakly relevant, or irrelevant, to the TE, while $r(x)$ carries the residual effect signal; thus directly targeting the differenced quantity is more efficient, favoring CEPIG-$\tau$.
> > >
> > > Within this cancellation family, we vary $\rho=\frac{k_{pred}}{k_{prog}+k_{pred}}$, analogous to the control used in AFA. Increasing $\rho$ does not simply add variables; it increases the share of effect-specific structure relative to shared/prognostic structure. Hence, within the cancellation family, the relative preference shifts toward CEPIG-$\tau$ as $\rho$ increases.
> > >
> > > **[L1]** To characterize when CEPIG becomes sub-optimal, we evaluate *one-step acq quality*: for each candidate point, we compare its acquisition score with its realized one-step utility (the reduction in test $\sqrt{PEHE}$ after adding that point and retraining once). We report: 1. Spearman score–utility correlation, 2. Top-1 utility, 3. Top-1 regret, with a random baseline for utility/regret.
> > >
> > > *1. Low data alone does not imply failure.* In Hahn dataset, we vary the warm-start size. Calibration deteriorates sharply at tiny warm starts (poor coverage, high NLL; Fig. 2), but acq quality does not collapse uniformly: across many moderate low-data warm starts, CEPIG-$\mu$ still achieves competitive or better Top-1 utility/regret than $\mu$-BALD (Fig. 3).
> > >
> > > *2. When joint structure is weak, the extra advantage shrinks.* We construct a setting with nearly independent PO surfaces. The extra gain from exploiting joint structure is small: the acquisition rules become much less distinguishable, with CEPIG-$\mu$ retaining only a modest edge, while the advantage of CEPIG-$\tau$ largely disappears (Fig. 4).
> > >
> > > *3. Severe structural misspecification degrades ranking quality for the compared rules.* Finally, we construct two harder datasets (Figs.5,6) whose structure is deliberately difficult for the model to represent, making effect-relevant joint dependence hard to capture accurately. In these regimes, score–utility correlation becomes weak or negative and practical ranking quality largely collapses, with Top-1 utility/regret moving close to random or worse. Thus, the key failure mode is severe structural misspecification, not low data alone.
> > >
> > > We thank the reviewer again and will improve the paper accordingly.

---

### Official Review · Reviewer_Wyev · 2026-03-11

**Soundness:** 4
**Presentation:** 4
**Significance:** 3
**Originality:** 3
**Overall Recommendation:** 4
**Confidence:** 3

**Summary:**

This paper addresses the mismatch in conventional strategies for CATE estimation via the principle of causal objective alignment. The authors propose a framework called Causal-EPIG to quantify uncertainty reduction in causal quantities, specifically focusing on aligning the acquisition target with unobserved causal quantities. Additionally, this paper demonstrates the compatibility of this framework with Gaussian Processes and Bayesian Causal Forests, providing theoretical bounds based on transductive learning. Extensive experiments show that this framework significantly outperforms the baseline in various settings.

The primary contributions include:
The authors point out the mismatch between acquisition targets of standard CATE estimation strategies and the actual goals of causal inference, proposing the Causal Objective Alignment principle to resolve this issue.
Providing rigorous proofs for convergence bounds.

**Compliance With Llm Reviewing Policy:**

Affirmed.

**Final Justification:**

I will keep my score.

**Key Questions For Authors:**

Since the normality tests are often rejected, how large is the error in the mutual information calculation when the BCF posterior is highly skewed?
The author point out that analyzing the convergence rate of the mean-marginal(MM-ITL) strategy requires a very strong assumption, which usually fails in the transductive setting. However, completely abandoning the theoretical analysis for the algorithm is unsatisfactory. Have the author considered providing a relaxed convergence bound to strengthen the paper?

**Limitations:**

YES

**Strengths And Weaknesses:**

# Strengths

This paper provide clear convergence bounds and build up solid theoretical results.
The framework offers a flexible design, introducing both mu and tau strategies that can be adapted depending on the specific task requirements.

# Weaknesses
The closed form mutual information relies on the assumption that the joint predictive distribution is Gaussian. Since real posteriors are often skewed, the mutual information calculation will be wrong if this assumption fails.
The theoretical guarantees only apply to the global formulation, which is too slow to use. The experiments use a faster mean-marginal formulation, but there is no theoretical proof for this approximated version.

---

> ### Author Rebuttal · Authors · 2026-03-30
>
> We are grateful for your careful reading and thoughtful suggestions, which will certainly strengthen the final paper. We address your specific questions in turn below.
>
> **[Q1.]** We agree that for BCF the Gaussian mutual information (MI) is an approximation rather than exact, since the predictive posterior is not Gaussian. The key question is therefore the size of the induced error and its impact on acquisition decisions.
>
> We evaluate this using posterior samples on Hahn nonlinear benchmarks by comparing moment-matched Gaussian MI with a nonparametric reference (kNN MI for the $\tau$-line), measuring both MI error and downstream decision quality.
>
> In the most skewed regime (nonlinear shift, budget 50), normality is strongly rejected (target rejection rate $0.9875\pm0.0125$, median $|\mathrm{skew}(\tau)|=0.91\pm0.16$). Despite this, the MI error remains small (absolute error $0.0153\pm0.0009$), and the decision error is smaller: the Gaussian top-1 candidate has regret $0.0049\pm0.0030$ under the reference MI. At budget 150, these further reduce to $0.0130\pm0.0008$ (MI error) and $0.0015\pm0.0011$ (regret).
>
> In terms of utility, the Gaussian approximation is not worse than the nonparametric reference in this high-skew regime: top-1 utility $0.0058\pm0.0723$ vs. $-0.0527\pm0.0617$.
>
> Thus, while normality is often rejected, the practical impact of the Gaussian approximation is limited: it yields small MI error and negligible ranking error in all regimes studied, including highly skewed posteriors. This is consistent with acquisition depending mainly on relative ranking, which is less sensitive to higher-order deviations (e.g., skewness) than to the dominant covariance structure captured by the Gaussian approximation.
>
>
>
> **[Q2.]** We clarify that our theory does not provide a direct convergence-rate comparison with misaligned objectives, but instead gives a layered justification of causally aligned acquisition by characterizing convergence of aligned target uncertainty and explaining efficiency differences. Importantly, aligned acquisition directly controls uncertainty in the target CATE, whereas misaligned objectives reduce uncertainty in surrogate quantities, leading to slower effective convergence in the estimand of interest.
>
> **(1) Interpretation of Thm. 4.4.** Thm. 4.4 establishes convergence of the target uncertainty (posterior uncertainty of potential outcomes / CATE) under the global aligned objective:
>
> $\mathrm{Var}[f(\tilde{x}^{\ast}) \mid \mathcal{D}\_{T}] \le \eta^{2}\_{\mathcal{D}\_{P}}(\tilde{x}^{\ast}) + C \frac{\gamma\_{n\_{T}}}{\sqrt{n\_{T}}}$.
>
> Its role is to justify greedy aligned acquisition by showing that it directly reduces uncertainty in the target causal estimand at a rate controlled by transductive information capacity, rather than providing general submodular optimality or direct comparison with all baselines. This yields a convergence guarantee for the aligned objective, not a universal optimality or cross-objective rate comparison.
>
> **(2) Scope of submodularity.** We have investigated this point. We do not find that Causal-EPIG-$\mu$-G is generally submodular, and therefore do not assume or claim universal submodularity. Following TAL, we instead identify regimes where approximate submodularity holds: it holds in regular regimes but may fail in transductive / distribution-shift settings due to dependence and synergy. This is further exacerbated in CATE estimation, where partial observability (one factual outcome per query) induces additional dependence. Consistent with [1], conditional mutual information objectives in such settings generally do not admit naive submodularity. We revise the paper to clarify when submodularity holds or breaks.
>
> **(3) MM formulation.** In response to this concern, we provide a relaxed TAL-style analysis for the practical MM strategy rather than abandoning theory. For the $\tau$-based case:
> $U_{\mathrm{MM}}(x,t) = \frac{1}{|X_{\mathrm{tar}}|} \sum_{x'} I(y;\tau(x') \mid x', \mathcal{D}_T),$
>
> which yields
>
> $\frac{1}{|X\_{\mathrm{tar}}|} \sum\_{x'} \mathrm{Var}[\tau(x') \mid \mathcal{D}\_T] \le \frac{1}{|X\_{\mathrm{tar}}|} \sum\_{x'} \eta^2(x') + C \frac{\bar{\gamma}\_n}{\sqrt{n}}.$
>
> This is a weaker counterpart of the global result but provides nontrivial theoretical support for the practical algorithm. For $\mu$-based MM, marginal uncertainty control similarly upper-bounds CATE uncertainty.
>
> **(4) Relation to naive / misaligned methods.** We do not claim misaligned acquisition is biased; under standard assumptions it remains consistent. The difference is efficiency and alignment: misaligned objectives (e.g., factual or parameter uncertainty) do not directly control uncertainty in the causal estimand and can lead to slower convergence in the target quantity. As shown in App (Prop. F.5), parameter-based criteria include components not directly useful for prediction, reducing sample efficiency without introducing bias.

---

> > ### Author Rebuttal · Reviewer_Wyev · 2026-04-04
> >
> > My concerns have been adequately addressed, I will maintain the score.

---

### Official Review · Reviewer_DCfA · 2026-03-13

**Soundness:** 3
**Presentation:** 2
**Significance:** 3
**Originality:** 2
**Overall Recommendation:** 4
**Confidence:** 2

**Summary:**

This paper considers CATE estimation through the lens of active learning. When the cost of acquiring samples is high, it is important to acquire samples efficiently to minimize the uncertainty in the final causal estimand. Existing method suffers from the objective misalignment as they target to reduce uncertainty in observed outcome instead of unobserved counterfactuals. The authors propose the Causal-EPIG acquisition function, which directly maximizes the mutual information gain with respect to the potential outcomes or the CATE estimand during each iteration. Mutual information can be easily calculated from Gaussian process models (or Gaussian approximations of Bayesian models).

**Compliance With Llm Reviewing Policy:**

Affirmed.

**Final Justification:**

Although novelty seems limited but this paper would still provided theoretical contribution to this line of work.

**Key Questions For Authors:**

- How does the convergence rate of the proposed method compare to the convergence rate of the naive method (with the objective misalignment). Does the naive framework lead to biases?

- In practice, how should practitioners choose step size (how many samples to acquire each time) under cost constraints? And is there a stopping criterion/threshold to terminate the acquisition process?

**Limitations:**

The authors discusses some limitations, but I would like see more discussion on the reliance on bayesian models as the CATE estimator.

**Strengths And Weaknesses:**

**Strengths**:
- The authors provided two frameworks that can target different data generating process and discusses when to choose between them.
- Provided theoretical analysis on the convergence rate of the proposed method.
- Simulation results demonstrate the superior performance of the proposed methods over baselines.


**Weaknesses**:
- There is limited novelty as the method builds on top of EIG framework.
- There isn't much discussion on the convergence rate result.
- The mutual information calculation seems to require the bayesian estimators. Can the model be adapted to other estimators for CATE as well?

**Others**:
- The notation of the subscript of $s$ and $s+1$ seems to not be defined in proposition 4.1

---

> ### Author Rebuttal · Authors · 2026-03-30
>
> We sincerely appreciate the reviewer’s constructive comments. We will ensure these refinements are integrated into the revised manuscript.
>
> **[W1]** We agree that our method builds on the EIG and does not introduce a new estimator; our contribution is to extend information-theoretic acquisition to directly target causal estimands rather than model parameters or factual outcomes.
>
> The contribution is a change in the acquisition target. Causal-BALD reduces parameter uncertainty, whereas Causal-EPIG directly reduces uncertainty in the target CATE, leading to different acquisition rankings and queried data. This is not merely conceptual; it leads to measurable differences in acquisition behavior and sample efficiency, as shown in our controlled experiments.
>
> This difference is not solely due to distribution shift. Active CATE estimation is inherently transductive: the target covariates define where the CATE must be accurate. Prediction-based acquisition can therefore be advantageous even without marginal shift, as it directly targets uncertainty in the target CATE rather than global parameter uncertainty.
>
> To isolate this effect, we conduct a controlled ablation comparing Causal-EPIG-$\mu$ and $\mu$-BALD in CMGP. The source pool is fixed as $X_{\mathrm{src}}\sim\mathcal{N}(0,I)$, while the target distribution is obtained by shifting only the treatment-effect-relevant coordinate ($\tau\_{\mathrm{direct}}$) and keeping the remaining coordinates unchanged; the shift strength is calibrated by KL divergence via $\delta=\sqrt{2KL}$.
>
> Across all metrics (AUC, PEHE), Causal-EPIG consistently outperforms $\mu$-BALD:
> 1. at zero shift, a clear gap already exists (final PEHE $\Delta \approx 0.36$);
> 2. under moderate shift ($KL\approx0.09$), the gap increases (final $\Delta \approx 0.43$);
> 3. the effect is non-monotonic over $KL\in[0,0.12]$, indicating that larger shift does not necessarily increase the advantage.
>
> Thus, the improvement is fundamentally due to target-aligned (transductive) acquisition, with distribution shift acting as a secondary factor that modulates this gap. Correspondingly, the differing theoretical properties are not merely a consequence of using a different evaluation criterion, but reflect that the two methods optimize different inferential targets and therefore induce genuinely different acquisition behavior.
>
> **[W2]** See Q2 of Reviewer Wyev.
>
> **[W3&L1]** We agree that exact MI computation relies on predictive uncertainty, naturally available in Bayesian models. However, our method does not require Bayesian inference per se, but only access to predictive samples (including joint uncertainty over target quantities), which provide the dependence structure needed for MI estimation without assuming an explicit parametric posterior.
>
> To show this, we implement a fully frequentist black-box extension using a bootstrap two-head TARNet, where uncertainty is obtained purely from bootstrap predictive samples without posterior inference, and MI is estimated via Monte Carlo.
>
> We evaluate this on all four Hahn benchmarks, pooling results across 20 acquisition trajectories. Causal-EPIG remains effective and consistently outperforms BALD: Causal-EPIG-$\mu$ wins against $\mu$-BALD on $75.0\%\pm5.2\%$ of acquisition points with $16.7\%\pm4.4\%$ relative improvement, while Causal-EPIG-$\tau$ wins against $\tau$-BALD on $85.0\%\pm5.1\%$ with $27.4\%\pm6.3\%$ improvement. Both EPIG variants also outperform random sampling, whereas BALD does not.
>
> These results indicate that the key requirement is not Bayesian estimation but access to reasonably calibrated predictive samples. When such samples are available, Causal-EPIG can be directly applied to flexible black-box CATE estimators and remains effective in practice, with joint uncertainty approximated directly from bootstrap samples.
>
> **[W4]** $s$ denotes the acquisition step. $\mathcal{D}\_s$ and $\mathcal{D}\_{s+1}$ represent the dataset before and after adding new sample.
>
> **[Q2]** The step size reflects a trade-off between adaptivity and computational cost: smaller batches improve label efficiency via more frequent updates, while larger batches reduce retraining overhead. Our ablation confirms this: smaller batches are more sample-efficient, whereas larger batches offer better runtime. This trade-off is more pronounced here, as each step evaluates information gain over the target set, making smaller batches preferable when precise target alignment is critical.
>
> Standard practice is to fix a label budget, as in our experiments. For early stopping, one can monitor reduction in target uncertainty, e.g., when acquisition utility plateaus or the average posterior uncertainty of $\tau(x)$ over the target set stabilizes over several iterations (e.g., negligible relative change across recent rounds), both indicating diminishing returns. Since our objective directly targets CATE uncertainty, this provides a principled stopping signal aligned with the final task.

---

> > ### Author Rebuttal · Reviewer_DCfA · 2026-04-03
> >
> > I thank the authors for addressing my questions, I will keep my score.

---

### Official Review · Reviewer_Y2g5 · 2026-03-13

**Soundness:** 3
**Presentation:** 3
**Significance:** 2
**Originality:** 1
**Overall Recommendation:** 3
**Confidence:** 4

**Summary:**

This paper proposes Causal-EPIG, an active learning framework for Conditional Average Treatment Effect (CATE) estimation. Compared to prior work on active learning for CATE, their acquisition function is more directly related to the causal estimand, i.e. CATE. Some theoretical and simulation results show that the proposed method outperform prior work such as CATE-EIG and CATE-BALD.

**Compliance With Llm Reviewing Policy:**

Affirmed.

**Final Justification:**

This paper proposes a target-aligned acquisition objective for causal active learning, replacing standard EIG with a criterion focused on CATE over a target population. The motivation is clear, and the empirical results support its usefulness, especially under distribution shift. The paper is well presented, and the rebuttal helps clarify implementation details, particularly the use of predictive sampling.

However, my overall evaluation remains largely unchanged. The work is sound, but in terms of originality and significance, it mainly introduces a modified objective rather than a new algorithm or theoretical advance. Key concerns from my original review, such as reliance on uncertainty estimation and assumptions about the target distribution, are only partially addressed and limit general applicability.

Overall, I view this as a reasonable but incremental contribution, and the rebuttal did not sufficiently change my evaluation. I therefore maintain a weak reject recommendation.

**Key Questions For Authors:**

1. When the CATE can be parameterized by a finite-dimensional parameter vector (e.g., $\tau(x) = \phi(x)^T \theta_\tau$), the two objectives appear closely related and may differ mainly by a population-dependent weighting over parameter directions. Could the authors clarify in which settings the proposed objective leads to fundamentally different acquisition behavior compared to Causal-EIG?
2. Because the acquisition objective averages information gain across many target covariate points, it may be significantly more expensive to compute than parameter-based criteria such as Causal-EIG. Could the authors provide a clearer comparison of computational complexity and runtime in practice, and discuss the trade-offs between computational cost and performance improvement?

**Limitations:**

yes

**Strengths And Weaknesses:**

Strength

The paper proposes a conceptually well-motivated acquisition objective that directly targets uncertainty in the CATE, aligning the active learning criterion with the causal estimand rather than model parameters or outcome predictions in prior work. This perspective is intuitive and potentially useful for causal active learning problems. Empirical results also suggest that the proposed acquisition rule can improve sample efficiency relative to several existing baselines in the simulation studies.

Weaknesses

1. The primary contribution is the proposal of a new information-gain objective targeting uncertainty in the CATE. The paper does not introduce a new learning algorithm or estimation procedure. When compared to prior works, the derived theoretical property is mainly because the evaluation criteria are different from previous papers.
2. The criteria used in prior works such as CATE-EIG and CATE-BALD seem to be also reasonable. So, the authors should not overstate the superiority of CATE-EPIG. For example, When the CATE is parameterized by a finite-dimensional parameter vector, e.g. nonparametric spline, the proposed objective may effectively reduce to a weighted form of parameter information gain, making the practical difference between the methods less clear and ends up being comparing which metric is more appropriate and typically non-conclusive.
3. The proposed acquisition rule requires computing mutual information quantities involving $\tau(x)$. The experiments in the paper include GP-based and DeepGP models that allow posterior sampling for uncertainty estimation. It is unclear how easily CATE-EPIG can be implemented when the CATE is estimated using more flexible or black-box approaches (e.g., meta-learners), where posterior uncertainty and covariance structures may not be readily available.
4. In practice, some of the covariate values $x^*$ in the target population may lie in regions that are not well represented or unseen in the currently labeled training data. In such cases, the posterior estimates of $\tau(x^*)$  and the associated uncertainty rely heavily on model extrapolation. The paper does not discuss how the acquisition rule behaves under this situation or whether the information gain estimates remain reliable when the model has limited support from observed data.

---

> ### Author Rebuttal · Authors · 2026-03-30
>
> We thank the reviewer for the helpful feedback. We will incorporate the suggested clarifications and improvements in the final version. Due to space constraints, some related points are addressed jointly with other reviewers’ comments. We respond point-by-point below.
>
> **[W1]** See W.1 of Reviewer DCfA.
>
> **[W2,Q1]** We agree that in finite-dimensional parametric settings (e.g., $\tau(x)=\phi(x)^\top\theta_\tau$), the two objectives can be closely related, and may even collapse to a target-dependent reweighting of parameter information gain in special cases. We therefore do not claim universal superiority over Causal-EIG/Causal-BALD. The difference becomes fundamental beyond this restricted regime:
>
> - **Nonlinear / functional dependence:** when $\tau(x)$ is induced by nonlinear outcome surfaces or more complex latent structure, reducing uncertainty in parameters does not generally imply proportional reduction in uncertainty of $\tau(x)$.
> - **Target-distribution dependence:** our objective is defined on $\tau(X_{\mathrm{tar}})$ and explicitly aggregates over $p_{\mathrm{tar}}(x)$, so it prioritizes uncertainty reduction where the target population lies; parameter-based criteria need not do this.
> - **Different acquisition rankings:** parameter uncertainty may be reduced most in directions that matter little for the target estimand, whereas Causal-EPIG scores candidates by their effect on the target CATE predictions themselves.
>
> Thus, the two objectives may coincide in simple low-dimensional parametric cases, but they induce genuinely different acquisition behavior once the estimand is a nonlinear, target-weighted functional rather than a direct linear readout of parameters. This distinction is also reflected in our controlled experiments, where the two objectives lead to different acquisition rankings beyond simple parametric regimes.
>
> **[W3]** See W3 of Reviewer DCfA.
>
>
> **[W4]** We agree that limited-support target regions are a critical stress case, since CATE uncertainty there necessarily relies on extrapolation. To test this, we add a disjoint-support experiment, where the target distribution is moved outside the labeled support along the treatment-effect-relevant dimension. The severity of the support gap is quantified via nearest-neighbor distance in the shifted subspace.
>
> We find that degradation is gradual rather than catastrophic (we observe similar trends for the $\mu$ variant). In a moderate disjoint-support regime (mean gap $1.1470\pm0.0047$, $q90=1.8958\pm0.0103$), Causal-EPIG-$\tau$ remains decision-reliable: its top-1 utility is $0.0547\pm0.0542$, and its top-1 regret is essentially zero ($0.0001\pm0.0001$), indicating correct selection. In contrast, $\tau$-BALD exhibits negative score–utility correlation ($-0.2238\pm0.1641$) and large regret ($1.1159\pm1.1058$).
>
> This difference arises because Causal-EPIG directly targets uncertainty in $\tau(x^*)$ on the target set, whereas BALD prioritizes uncertainty in predictive quantities that may not align with the extrapolated treatment effect.
>
> In a more extreme no-support regime, both methods fail and top-1 regrets become large ($\approx 1.07$). Thus, we do not claim reliability under arbitrary extrapolation; rather, our results show that Causal-EPIG remains reliable deeper into the limited-support regime than the matched BALD baseline, before both methods eventually fail.
>
>
> **[Q2]** We agree that the global formulation is substantially more expensive: it requires joint covariance operations over the full target set and scales as $O(n_P\,n_{\mathrm{tar}}^3)$.
>
> In practice, however, we adopt the summation / mean-marginal formulation (App. E.3), whose total acquisition cost is $O(n_P\,n_{\mathrm{tar}}\,S)$, where $S$ is the number of posterior samples. This scales linearly in the pool and target sizes, and is one to two orders of magnitude faster than the global variant in our runtime analysis (e.g., Causal-EPIG-$\mu$: $0.45$s vs. $9.27$s).
>
> Compared to parameter-based criteria such as BALD/Causal-EIG, our method is still more expensive because it explicitly scores candidates by their effect on the target causal estimand over $X_{\mathrm{tar}}$, rather than optimizing a global parameter objective. In practice, this introduces an additional factor linear in $n_{\mathrm{tar}}$ compared to parameter-based criteria, which do not require evaluation over the target set. This is a deliberate computational-statistical trade-off: parameter-based methods are cheaper, while Causal-EPIG incurs additional cost to directly optimize uncertainty reduction in the target CATE. In the intended regimes of costly outcome acquisition, this overhead is often dominated by labeling cost, while the summation formulation keeps the method practically scalable.

---

> > ### Author Rebuttal · Reviewer_Y2g5 · 2026-04-02
> >
> > I thank the authors for the detailed and constructive rebuttal. The clarification that the method can be implemented using predictive samples (e.g., bootstrap with TARNet), without requiring fully Bayesian inference, partially addresses my original concern regarding applicability beyond Bayesian models. The additional experiment under zero distribution shift is also helpful in clarifying when gains arise.
> >
> > However, my main concerns remain only partially resolved and correspond directly to points raised in my original review:
> >
> > 1. As discussed in my review, the paper primarily introduces a new acquisition objective built on EIG rather than a new learning or inference procedure. The rebuttal confirms this positioning. While the objective is well-motivated, the contribution remains largely a reformulation, and it is still unclear how broadly impactful this change is beyond specific regimes, while it is unclear how to tell if a real application is under such regimes or not.
> >
> > 2. My original review questioned how the method behaves when evaluating target covariates outside the support of labeled data. This issue remains: acquisition decisions depend on uncertainty estimates in regions with limited data, where extrapolation may be unreliable.
> >
> > 3. As noted previously, the framework assumes access to a target covariate distribution. The rebuttal does not fully clarify how this distribution is obtained in practice, especially in active learning settings where sampling is non-random and may bias empirical estimates.
> >
> > 4. I previously raised concerns about computational overhead. While approximations are discussed, the method still requires evaluating acquisition scores over target samples and estimating information gain via sampling, which may be nontrivial in larger-scale or more flexible modeling settings.
> >
> > Overall, while the rebuttal improves clarity and addresses some implementation concerns, the remaining issues are consistent with my original review and relate to the core novelty, robustness of uncertainty estimation, and practical assumptions, which would require more substantial revisions to fully resolve.

---

### Decision · Program_Chairs · 2026-04-30

**Decision:**

Accept (regular)

**Comment:**

While the reviewers mentioned concerns with the technical novelty, the reviewers found this paper addresses a gap in the literature by applying EPIG, an active learning algorithm that more directly targets the objective, to CATE estimation in causal active learning. The paper shows this method has strong experimental performance in simulation, and theoretically connects Causal-EPIG to the CATE objective. Given this contribution, I recommend acceptance.